# Birefringent Fourier filtering for single molecule coordinate and height super-resolution imaging with dithering and orientation

Valentina Curcio[1,3], Luis A. Alemán-Castañeda [1,2,3], Thomas G. Brown[2], Sophie Brasselet [1✉] & Miguel A. Alonso [1,2✉]

Super-resolution imaging based on single molecule localization allows accessing nanometric-scale information in biological samples with high precision. However, complete measurements including molecule orientation are still challenging. Orientation is intrinsically coupled to position in microscopy imaging, and molecular wobbling during the image integration time can bias orientation measurements. Providing 3D molecular orientation and orientational fluctuations would offer new ways to assess the degree of alignment of protein structures, which cannot be monitored by pure localization. Here we demonstrate that by adding polarization control to phase control in the Fourier plane of the imaging path, all parameters can be determined unambiguously from single molecules: 3D spatial position, 3D orientation and wobbling or dithering angle. The method, applied to fluorescent labels attached to single actin filaments, provides precisions within tens of nanometers in position and few degrees in orientation.

[1] Aix Marseille Univ, CNRS, Centrale Marseille, Institut Fresnel, F-13013 Marseille, France. [2] The Institute of Optics, University of Rochester, Rochester, NY 14627, USA. [3]These authors contributed equally: Valentina Curcio, Luis A. Alemán-Castañeda. ✉email: sophie.brasselet@fresnel.fr; miguel.alonso@fresnel.fr

Biological functions in cells and tissues are driven by the molecular-scale organization of biomolecular assemblies, which arrange in precise structures that are essential, for instance, in biomechanics and morphogenesis. A way to assess such organization is to monitor the orientation of fluorescent labels, in conditions where the label is sufficiently rigidly attached to the biomolecule of interest[1–4]. Monitoring orientational behavior of fluorescent molecules is still a challenge, however, because both orientational fluctuations and mean orientation need to be quantified. In particular, measurements can be strongly biased by the fact that molecular orientations may fluctuate at a time scale faster than the measurement integration time, which occurs naturally in biological media even in fixed conditions[2,4,5]. Recent studies have aimed at adding orientation information to super-resolution imaging, which relies on single-molecule localization. Orientation and position are however difficult parameters to disentangle, leading to possible localization biases[6,7]. A single molecule's point spread function (PSF) is intrinsically altered by its orientational properties[4,6]. Several methods have capitalized on this property by using Fourier-plane phase modification of the PSF[7–10], or imaging finely sampled PSFs[11]. However, these approaches apply only to molecules with fixed position. Recent proposals to access the missing information on wobbling rely on adding complexity to the PSF via phase filtering[12] or by using the index mismatch sensitivity of the PSF's shape[4], although the axial component of the single molecules' 3D position remains inaccessible. Other approaches use defocused imaging[5,13], but they require either fixed orientations or pre-determined spatial localization of the molecules. Alternatively, it is possible to preserve less-altered PSF images and restrict the measurements to 2D in-plane orientations by working under relatively low numerical aperture conditions and splitting polarization components[2], or using sequential polarization illumination[14–17]. So far, none of these techniques have allowed the simultaneous measurement of 3D orientational properties (including both orientational fluctuations and mean orientation) and 3D spatial position of single molecules, in a single-shot image scheme compatible with super-resolution localization. The main challenge is that the axial position of single molecules and their 3D orientational fluctuations (e.g., their wobbling) are intrinsically coupled by the imaging techniques.

Here, we propose a simple method to engineer the molecular PSF so that it efficiently encodes information about all these properties with very little coupling. The method is based on Fourier-plane filtering not only in phase but also in polarization by using spatially varying birefringence. It builds upon a prior technique for single-shot imaging polarimetry[18–20], where polarization is encoded in the shape of the PSF. This approach has been applied to multiple scattering measurements[21] as well as to the polarimetric characterization of multicore fibers[22]. In this work, we show that the same operating principle can be used to retrieve significantly more degrees of freedom when applied to imaging fluorescing molecules, where the PSFs encode information not only of the molecules' transverse coordinates $(x, y)$ but also of their axial height $z$, and of the three-dimensional correlations of the emitted light, which translate into the orientation of the molecules, namely the azimuthal angle $\xi$, the polar angle $\theta$, and the state of wobbling or dithering characterized by the average cone solid angle $\Omega$ (Fig. 1a). Furthermore, we show that there is negligible coupling in the dependence of the PSFs on the relevant parameters being measured, that the technique involves almost no photon losses, and that transverse spatial resolution is high since the PSFs encoding this information are only about twice as large as those of diffraction-limited imaging. We refer to the method as coordinate and height super-resolution imaging with dithering and orientation (CHIDO).

## Results

**PSF encoding through a birefringent mask.** The basis of the proposed technique is the placement at the pupil plane of an element referred to as a stressed-engineered optic (SEO), which is a BK7 glass window subjected to forces with trigonal symmetry at its edges[18,19,23] (see "Methods"). The spatially varying birefringence pattern that naturally results in the vicinity of the force equilibrium point has been shown to be essentially optimal for applications in polarimetry, in the sense that it efficiently encodes polarization information in the PSF's shape while causing the smallest possible increase in PSF size[24]. This birefringence pattern is described by the following Jones matrix (in the linear polarization basis) in the Fourier plane of the detection path (Fig. 1b)

$$\mathbb{J}(u) = \cos\frac{cu}{2}\begin{pmatrix} 1 & 0 \\ 0 & 1 \end{pmatrix} + \mathrm{i}\,\sin\frac{cu}{2}\begin{pmatrix} \cos\varphi & -\sin\varphi \\ -\sin\varphi & -\cos\varphi \end{pmatrix},$$
(1)

where $(u, \varphi)$ are polar pupil coordinates normalized so that $u = 1$ corresponds to the pupil's edge, and $c$ is a coefficient that depends on the stress within the SEO and the radius of the pupil being used. This parameter can be chosen to optimize the system's performance: small $c$ keeps the extension of the PSFs more restricted, but reduces the amount of information they carry about orientation and $z$ displacement, while large $c$ has the opposite effect[18,24]. After passing through the SEO, the two circular polarization components are separated to form two images by inserting a quarter-wave plate (QWP) followed by a Wollaston prism (Fig. 1b). A Fourier-plane image under circularly polarized illumination shows the effect of the SEO's spatially varying birefringence as a Fourier mask on the two detection channels (Fig. 1c).

We now show that the combination of the SEO and the separation of the two circular polarization images allows encoding information about a molecule's orientation and axial displacement in the shape of the PSFs. Let us model the fluorescing molecule as a quasi-monochromatic point dipole that can have any orientation (fixed or fluctuating) in three dimensions[5,13,25–28]. For now, we assume that this dipole is at the center of the object focal plane of the objective, $(x, y, z) = (0, 0, 0)$; the effects of lateral and axial displacements will be discussed later. The dipole is placed in a homogeneous medium, at a distance to the glass coverslip larger than the wavelength. This source can be described by the $3 \times 3$ second moment (or correlation) matrix $\mathbf{\Gamma}$ with elements $\Gamma_{ij} = \langle E_i^* E_j \rangle$ with $i, j = x, y, z$, $E_i$ being the radiated field components, and the angular brackets denoting an average over the integration time of the detector[5] (Supplementary Note 1). This type of $3 \times 3$ correlation matrix has also been used to study nonparaxial polarization[29–34]. For the sake of analogy with standard polarimetry (where the correlation matrix is only $2 \times 2$), we write $\mathbf{\Gamma}$ in terms of the generalized 3D Stokes parameters $S_n$, which are the coefficients of the expansion of this matrix in terms of the Gell–Mann matrices $\mathbf{g}_n$ (instead of the Pauli matrices used for $2 \times 2$ correlations, whose coefficients are the standard Stokes parameters)[29]. The resulting expression is

$$\mathbf{\Gamma} = \sum_{n=0}^{8} S_n \mathbf{g}_n = \begin{pmatrix} \frac{S_0 + S_8}{\sqrt{3}} + S_1 & S_2 - \mathrm{i}S_3 & S_4 - \mathrm{i}S_5 \\ S_2 + \mathrm{i}S_3 & \frac{S_0 + S_8}{\sqrt{3}} - S_1 & S_6 - \mathrm{i}S_7 \\ S_4 + \mathrm{i}S_5 & S_6 + \mathrm{i}S_7 & \frac{S_0 - 2S_8}{\sqrt{3}} \end{pmatrix}.$$
(2)

Note that we use a nonstandard numbering scheme for the Gell–Mann matrices: the elements $n = 1, 2, 3$ are cycled so that the resulting parameters $S_n$ reduce to the standard Stokes parameters for $n = 1, 2, 3$ when the field's $z$ component vanishes.

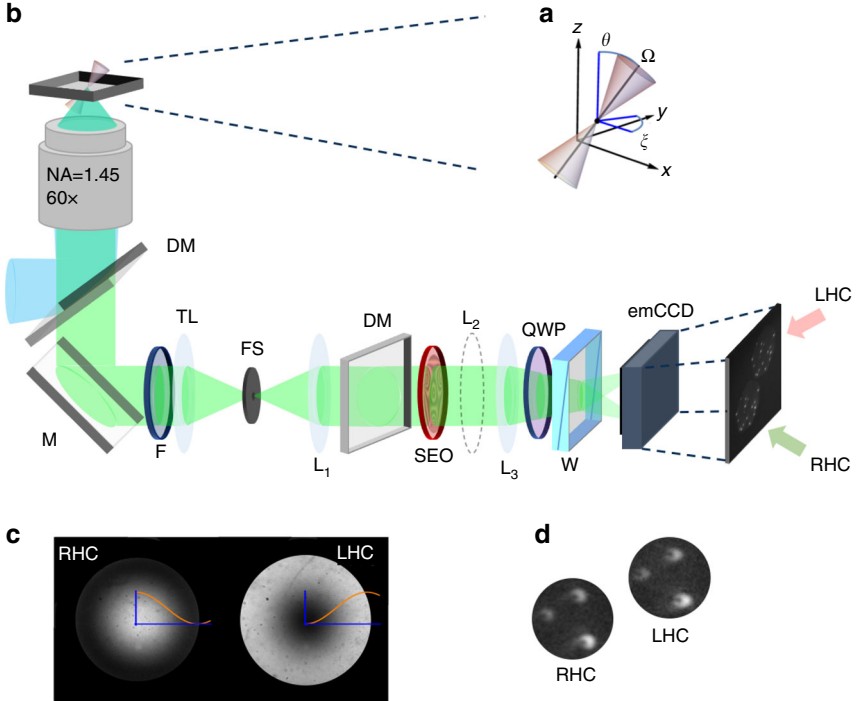

**Fig. 1 CHIDO imaging principle. a** Parameters defining the 3D position $(x, y, z)$, orientation $(\xi, \theta)$, and wobbling-subtended solid angle $\Omega$. **b** Optical setup (see "Methods"). DM dichroic mirror, M mirror, F fluorescence filter, TL tube lens, FS field stop. $L_1$, $L_2$, and $L_3$ lenses, QWP quarter-wave plate, W Quartz Wollaston polarizing beamsplitter, emCCD emCCD camera, RHC and LHC right-hand circular and left-hand circular polarized images. **c** Back-focal plane imaging of the SEO illuminated by a circular polarization (the sample is a homogeneous fluorescent sample, see "Methods"). **d** Direct image of isolated emitters (fluorescence beads, see Section "Methods") in the same polarized emission conditions.

(In this case, $S_0$ differs from the corresponding Stokes parameter for paraxial light by a factor of $\sqrt{3}/2$.)

Several measures have been proposed for the degree of polarization of nonparaxial light[33,34], one of them[30–32] having a definition in terms of the generalized 3D Stokes parameters that resembles the standard one for paraxial light

$$P_{3D} = \frac{1}{S_0}\left(\sum_{n=1}^{8} S_n^2\right)^{1/2} = \left[\frac{3\mathrm{tr}\boldsymbol{\Gamma}^2}{2(\mathrm{tr}\boldsymbol{\Gamma})^2} - \frac{1}{2}\right]^{1/2}. \tag{3}$$

In the present context, this degree of polarization is related to the amount of wobbling of the fluorescent dipole source. For a dipole wobbling uniformly within a cone, the cone solid angle $\Omega$ is a monotonic function of this degree of polarization (see Supplementary Note 2)

$$\Omega = \pi\left(3 - \sqrt{1 + 8P_{3D}}\right) \qquad \text{or} \qquad P_{3D} = \frac{(3\pi - \Omega)^2 - \pi^2}{8\pi^2}. \tag{4}$$

Note that for isotropic wobbling (i.e., when the two smallest eigenvalues of $\boldsymbol{\Gamma}$ are equal), $P_{3D}$ coincides with the rotational mobility parameter proposed to characterize wobbling[35]. The relation between these two measures is described in Supplementary Note 2; these and other related measures of polarization have simple geometric interpretations[34].

Let the PSFs at the two detector regions be denoted as $I^{(p)}$, where $p$ labels the polarization component being imaged at the corresponding detector: $p = \mathcal{R}$ for right-hand circular (RHC) and $p = \mathcal{L}$ for left-hand circular (LHC) (Fig. 1d). As shown in Supplementary Note 1, these PSFs depend linearly on the

generalized Stokes parameters according to

$$I^{(p)}(\boldsymbol{\rho}) = \sum_{n=0}^{8} S_n \mathcal{I}_n^{(p)}(\boldsymbol{\rho}), \tag{5}$$

where $\mathcal{I}_n^{(p)}$ are contributions to the PSF corresponding to each generalized Stokes parameter. Expressions for these contributions are derived in Supplementary Note 1, and theoretical images for some of them at $z = 0$ are shown in the top row of Fig. 2. Note that Fig. 2 does not include images for $\mathcal{I}_3^{(p)}$, $\mathcal{I}_5^{(p)}$, and $\mathcal{I}_7^{(p)}$ because they are not of interest to the current problem. (The complete set is shown in Supplementary Fig. 1.) This is because, as we can see from Eq. (2), the generalized Stokes parameters $S_3$, $S_5$, and $S_7$ correspond to the imaginary part of $\boldsymbol{\Gamma}$ and therefore encode information about the helicity of the emitted field, which is assumed not to exist since the emitters are (possibly wobbling) linear dipoles. Nevertheless, if the particles did emit light with some helicity, these elements could be incorporated into the treatment.

An important feature of the SEO's birefringence pattern is that it makes this set of PSF components nearly orthogonal while keeping their extension almost as small as possible (in analogy to the case of paraxial polarization[24]). This approximate orthogonality implies the strong decoupling of the information for each parameter, as will be discussed in the next section. Another desirable aspect of using the SEO as a filter at the pupil plane is the resulting approximate achromaticity over a spectral range corresponding to fluorescence spectral widths (typically 100 nm), in contrast to PSF-engineering methods based on pure phase masks. The only chromatic dependence of the Jones matrix in Eq. (1) is within the parameter $c$, which is roughly inversely proportional to the wavelength. This variation compensates the natural scaling of the PSF with wavelength, such that the PSFs resulting from the integration over the fluorescence spectrum

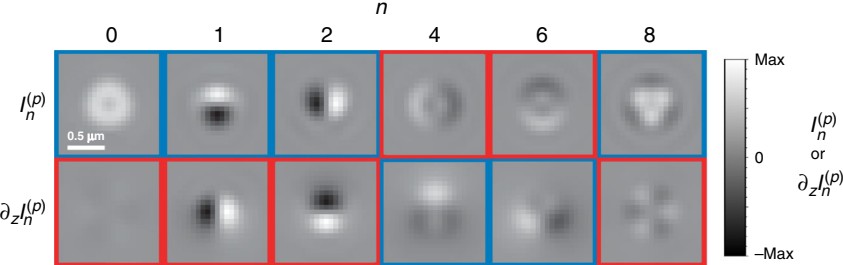

**Fig. 2 Theoretical PSF components in CHIDO imaging.** The figure shows both $\mathcal{I}_n^{(p)}$ and $\partial_z \mathcal{I}_n^{(p)}$ for $c = \pi$, $z = 0$, and $p = \mathcal{R}$. The corresponding components for $p = \mathcal{L}$ are identical, except that those surrounded by red boxes would have the opposite sign. Each row is normalized separately as their units are different.

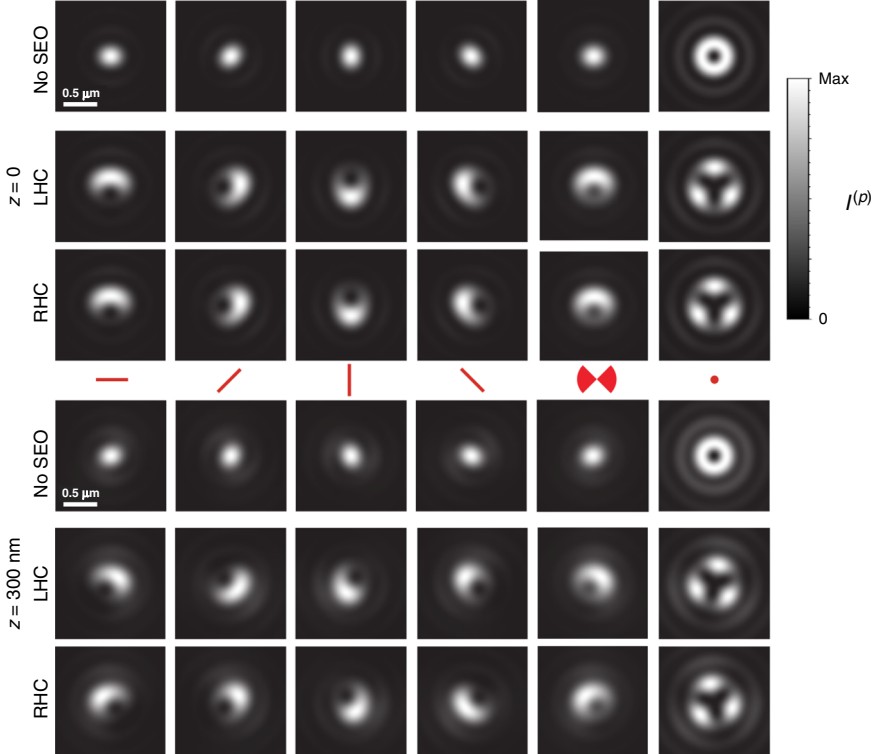

**Fig. 3 Theoretical PSFs formed from specific dipole orientation and wobbling.** The PSFs are shown at the nominal focal plane $z = 0$ (top) and at $z = 300$ nm (bottom), corresponding to five different dipole orientations: the first four from left to right correspond to nonwobbling dipoles within the $xy$ plane ($\theta = 90°$) and for $\xi = 0°$, $45°$, $90°$, and $135°$, respectively, while the sixth column corresponds to a nonwobbling dipole in the $z$ direction ($\theta = 0°$). The fifth column corresponds again to $\xi = 0°$, but for the dipole wobbling within an angle $\delta$ of $90°$ (corresponding to $\Omega = 0.6\pi$ and $P_{3D} = 0.6$). For both heights, the top row shows for reference the (diffraction-limited) PSFs without the SEO, while the rows labeled RHC and LHC show the two PSFs for CHIDO. Note from the first four columns that a rotation of the dipole within the $xy$ plane causes an approximate joint rotation of both PSFs in a direction opposite to that of the dipole and by twice the angle. A change in height, on the other hand, causes approximate rotations of both PSFs in opposite directions with respect to each other. Wobbling causes a blurring of the PSFs. PSF pairs for other orientations and wobbling angles are shown in Supplementary Movie 1.

being measured are nearly indistinguishable from those resulting from only the peak wavelength. If a measurement required larger wavelength ranges, appropriate recalibration must be used, as in any PSF-engineering-based technique. The chromatic dependence of CHIDO is discussed in Supplementary Note 1 and quantified in Supplementary Fig. 2.

When the emitter is within the plane conjugate to the image, each of its two images is a linear combination of the six PSFs shown in the top row of Fig. 2, according to Eq. (5). The possible differences between the two images arise from a global sign change of two members of the PSF basis set, $\mathcal{I}_4^{(p)}$ and $\mathcal{I}_6^{(p)}$. Figure 3 and Supplementary Movie 1 show simulations of measured PSF pairs corresponding to several dipole orientations.

Also shown in Fig. 3 for comparison are the corresponding diffraction-limited PSFs resulting from not using the SEO, whose shape is nearly independent of the in-plane angle $\xi$. In contrast, when the SEO is used, the PSFs acquire a crescent shape for a dipole within the $xy$ plane, and a rotation of the dipole within this plane results in an approximate rotation of both PSFs, in the opposite sense as the dipole and by twice the angle. Note that these PSFs are only about twice as large as the diffraction-limited ones. A dipole in the $z$ direction, on the other hand, corresponds to a PSF with trigonal symmetry (which is also only about twice as large as the corresponding diffraction-limited PSF). Wobbling of the dipole about its nominal direction has the effect of blurring the PSFs in a predictable way. Therefore, the parameters $S_n$ can be

estimated by making the superposition in Eq. (5) agree as closely as possible with the measured pair of PSFs (see Supplementary Note 2). From these parameters, the matrix $\Gamma$ can be constructed using Eq. (2), which is real and symmetric because $S_3 = S_5 = S_7 = 0$. The central direction of the dipole source is then estimated as that of the eigenvector of $\Gamma$ with the largest eigenvalue. The remaining eigenvectors and eigenvalues provide information about the wobbling of the molecule (Supplementary Note 2). In addition, in the minimization procedure that leads to the retrieval of the parameters $S_n$, the transverse $x$, $y$ position of each emitter can be estimated to within a fraction of a pixel (Supplementary Note 2). This analysis can be performed simultaneously for multiple emitters within an image, as long as their PSFs do not overlap.

In addition to orientation and transverse localization, the measured images provide information about axial localization, since the PSFs depend on $z$ (significantly more so than those without the SEO). As shown in Fig. 3 and Supplementary Movie 1, a variation in $z$ for a dipole oriented within the $xy$ plane causes a rotation of both measured PSFs, but these rotate in opposite directions. This is in contrast with an in-plane rotation of the dipole (a change in $\xi$), which causes common rotation of the PSFs. Therefore, if only the image corresponding to one polarization component were used, it would be nearly impossible to distinguish height from orientation, but imaging separately both circular components fully decouples $z$ and $\xi$. A rotation of the two PSFs in opposite directions also occurs when a dipole oriented in the $z$ direction changes height. To provide intuition for this behavior, the bottom row of Fig. 2 shows $\partial_z \mathcal{I}_n^{(p)}$, namely the derivative with respect to $z$ of each of the basis elements at the plane $z = 0$. The similarity of $\partial_z \mathcal{I}_1^{(p)}$ and $\partial_z \mathcal{I}_2^{(p)}$ with $\mathcal{I}_2^{(p)}$ and $\mathcal{I}_1^{(p)}$, respectively, explains the fact that both the in-plane rotation and vertical displacement of a horizontal dipole cause rotations; the distinguishability between them arises because $\partial_z \mathcal{I}_1^{(p)}$ and $\partial_z \mathcal{I}_2^{(p)}$ have opposite signs for $p = \mathcal{R}$ and $\mathcal{L}$, while $\mathcal{I}_2^{(p)}$ and $\mathcal{I}_1^{(p)}$ do not, making in-plane orientation and height decoupled in the retrieval process.

**Cramér–Rao analysis.** In order to estimate the sensitivity of CHIDO, we use Cramér–Rao (CR) lower bounds[36,37] on the uncertainties of the six parameters being measured ($x$, $y$, $z$, $\xi$, $\theta$, $\Omega$). These bounds were deduced from a numerical calculation of the inverse of the Fisher Information matrix, in this case of dimension $6 \times 6$. Each of the six lower bounds depends on all six parameters, as well as on the photon number, the SEO's stress parameter $c$, the pixelation of the PSFs, and the signal-to-background ratio, namely the ratio of the PSF peak intensity to the uniform illumination background. To reduce the size of the parameter space being explored, we fix $c = 1.2\pi$ and assume a pixelation level comparable to that of our experimental implementation.

Let us start by considering the CR lower bounds for the standard deviations of the directional parameters $\xi$, $\theta$, and $\Omega$. Supplementary Note 3 presents the derivation of the simple order-of-magnitude estimates of these bounds, based on the near orthogonality of the PSF components, which permits the approximation of the diagonal terms of the Fisher matrix

$$\sigma_\theta \approx \frac{2}{P_{3D}\sqrt{6\widetilde{\mathcal{N}}}} = \frac{(4\pi)^2}{(8\pi^2 - 6\pi\Omega + \Omega^2)\sqrt{6\widetilde{\mathcal{N}}}}, \qquad (6a)$$

$$\sigma_\xi \approx \frac{\sigma_\theta}{\sin\theta}, \qquad (6b)$$

$$\sigma_\Omega = \frac{\sigma_{P_{3D}}}{\partial_\Omega P_{3D}} \approx \frac{1.43}{\partial_\Omega P_{3D}\sqrt{\widetilde{\mathcal{N}}}} = \frac{1.43(4\pi^2)}{(3\pi - \Omega)\sqrt{\widetilde{\mathcal{N}}}}, \qquad (6c)$$

where $\widetilde{\mathcal{N}} = \mathcal{N}/(1 + 2\,\text{SBR}^{-1})$, with $\mathcal{N}$ being the number of signal photons. Here, $\sigma_\xi$ and $\sigma_\theta$ are given in radians and $\sigma_\Omega$ is given in steradians. The simple dependence of these bounds on $\theta$ and $P_{3D}$ suggests a definition for a global measure of directional/wobbling precision as $\sigma_{\text{Dir}} = P_{3D}^2 \sin\theta\,\sigma_{P_{3D}}\sigma_\theta\sigma_\xi \approx \widetilde{\mathcal{N}}^{-3/2}$, which depends solely on the number of photons and SBR values. Interestingly, if we were to consider a spherical space where $P_{3D}$ is the radial variable and $\xi$ and $\theta$ are the azimuthal and polar angles, this measure would correspond to the volume element in this space implied by the CR bounds. The value of $\widetilde{\mathcal{N}}^{3/2}\sigma_{\text{Dir}}$ was calculated numerically for 10,000 randomly selected cases with inverse SBR between 0 and 3, heights between $-200$ and 200 nm, and the orientational parameters covering uniformly the sphere with coordinates ($P_{3D}$, $\xi$, $\theta$); as shown in Supplemental Fig. 3b, the resulting distribution indeed peaks near unity.

Figure 4 shows comparisons of the simple estimates in Eq. (6c) (dotted lines) to numerically calculated values (solid lines) of the CR lower bounds for several values of the parameters, both in the absence of background photons (a–d), and for a SBR of one-third representative of the single-molecule measurements presented later (e–h). The agreement between the approximate expressions and the more rigorous theoretical calculations is very good as expected. The figure also shows the CR lower bounds for the three spatial parameters. For the sake of illustration, these results assume a total of 10,000 signal photons over the two detection channels; the obtained levels of error scale as the inverse of the square root of the photon count. Figure 4a, e show the variations with $\xi$ of the six CR lower bounds for a nonwobbling fluorophore within the plane $z = 0$ with in-plane orientation ($\theta = \pi/2$). Note that, indeed, the dependence of the lower bounds on the in-plane orientation of the fluorophore is not very significant. The variation of the CR bounds as the off-plane orientation changes is illustrated in Fig. 4b, f (assuming $\xi = 0$), showing that this change of orientation makes the uncertainty in $z$ first decrease slightly and then increase by less than a factor of two. The uncertainty in $\xi$ grows as the inverse of $\sin\theta$, as expected. Figure 4c, g show that even moderate amounts of wobble have an adverse effect on the CR bounds for height and direction. These lower bounds are indeed roughly multiplied by three when $\Omega$ reaches $\pi$. Finally, given that the PSFs occupy a sufficiently large number of pixels, the CR bounds depend very weakly on changes in $x$ and $y$. The dependence on $z$ is more significant, as can be seen in Fig. 4d, h, which show that 3D spatial localization is affected, given the expansion of the PSFs with defocusing; the different behavior in $x$ and $y$ is due to the chosen molecule orientation ($\xi = 0$). On the other hand, the effect of $z$ over the level of precision of the determination of direction is lower, particularly for low background.

The estimates just shown, as well as other simulations we performed, indicate that when a few thousand photons are measured, one can expect a precision in transverse position of a few nanometers, and an uncertainty in $z$ about three or four times larger. The corresponding precision in the determination of orientation angles is of a few degrees, and for wobble, it is on the order of tenths to hundredths of sterradians. These levels of precision are comparable or superior to those of other approaches restricted to the estimation of a subset of the parameters, whether they are based on engineering the PSF[8,9,12] or on observing the natural change in shape of unengineered PSFs[4,13]. As shown in Supplementary Note 2, these estimations are not only precise but

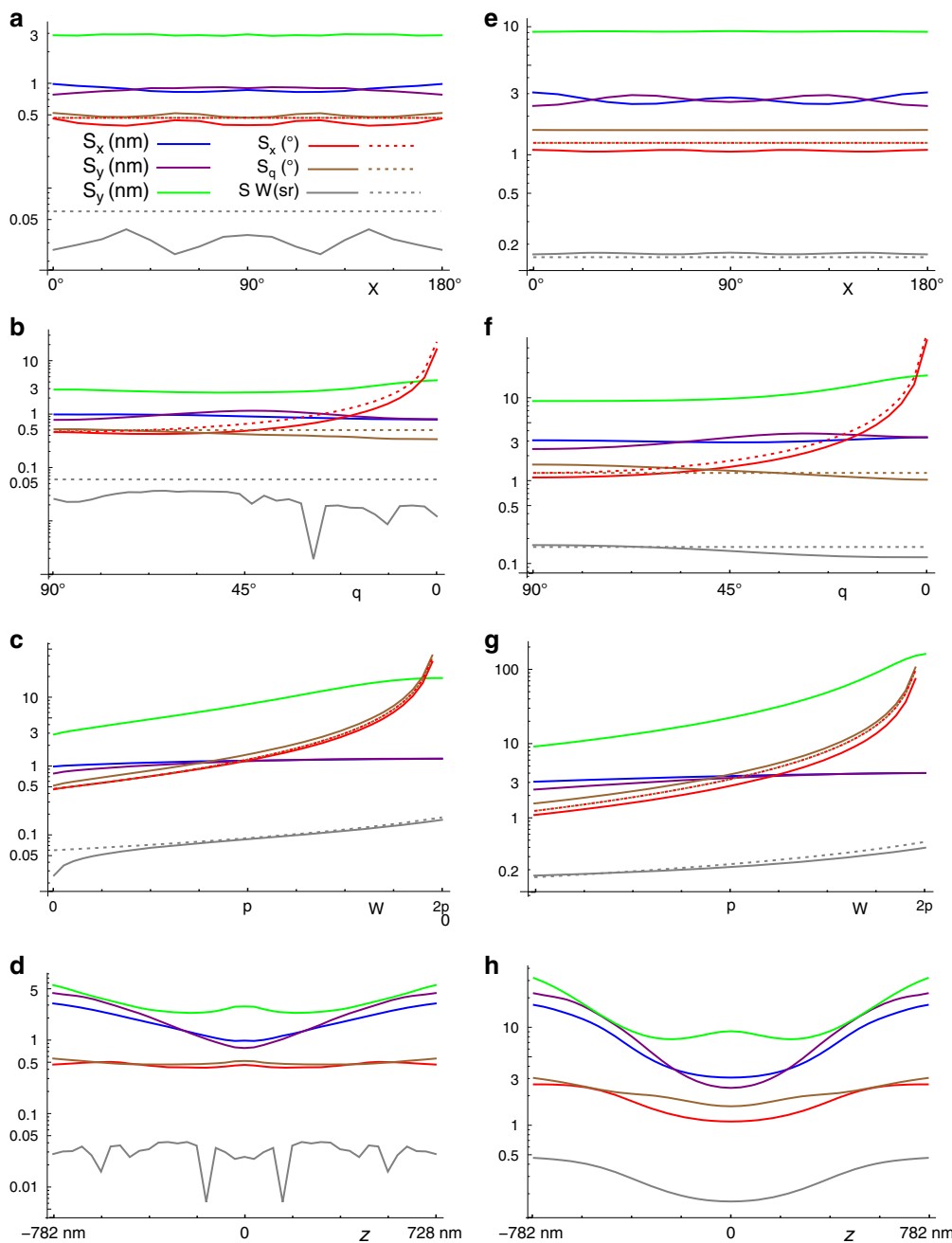

**Fig. 4 Cramer–Rao lower bounds for the six measured parameters.** These plots assume 10,000 signal photons over both channels, for **a–d** no background photons, and **e–h** a SBR of 1/3, corresponding to a background of about 250 photons per pixel. The parameters are **a**, **e** $x = y = z = 0$, $\theta = 90°$, $\Omega = 0$, and varying $\xi$; **b**, **f** $x = y = z = 0$, $\xi = 0°$, $\Omega = 0$, and varying $\theta$; **c**, **g** $x = y = z = 0$, $\xi = 0°$, $\theta = 90°$, and varying $\Omega$; **d**, **g** $x = y = 0$, $\xi = 0°$, $\theta = 90°$, $\Omega = 0$, and varying $z$. The units for each curve are indicated in the legend in (**a**). The dashed lines indicate the simple estimates in Eq. (6c).

they also involve relatively low levels of coupling between parameters. These results were validated by performing Monte Carlo simulations on randomly generated data with similar signal and background levels (Supplementary Note 3). The retrievals were based on the maximization of the normalized correlation with the model PSFs, which was chosen due to its simplicity and speed. The resulting standard deviations were found to be reasonably close to the CR lower bounds (about 2–5 times larger, as can be seen in Supplementary Fig. 6), despite the simplicity of the retrieval method. Importantly, no systematic bias in the retrieved parameters was found, even in the presence of background, except near the endpoints of the allowed range of

values of $\Omega$, which is natural given the finite, nonperiodic nature of this parameter (Supplementary Figs. 6 and 7).

Finally, to evaluate the robustness of the method with respect to image aberrations, the CR calculations were repeated assuming that the system presents one wave of spherical aberration. While this aberration does change the shape of the PSFs, the CR bounds remain largely the same, as shown in Supplementary Fig. 5. Note, however, that the presence of aberrations, if not accounted for in the model used for the retrieval of the parameters, will likely introduce (nonuniform) bias in the results, as would be the case for any other super-resolution-based method.

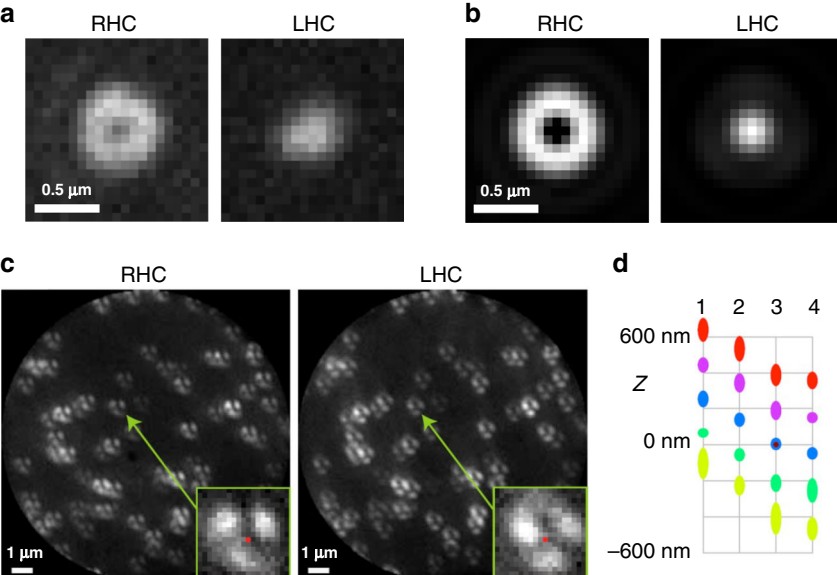

**Fig. 5 Experimental PSFs obtained from nanobeads under circular polarization and simulated z-oriented dipole. a** Measured and **b** simulated images for two nanobeads followed by a circular polarizer. **c** Image pair for a group of nanobeads with a S waveplate inserted at the pupil plane for simulating emitters oriented in the z direction. For these images, the integration time is typically 1 s (camera gain 300). The insets show zooms of the PSFs of a particular bead, where the red dot indicates the retrieved (x, y) coordinates. **d** Estimation of z (average—center of the ellipse, and standard deviation—height of the ellipse) for the five defocused measurements of the four sets of measurements. The brown dot at the central position for set 3 indicates that it corresponds to the images shown in part (**c**). All retrieved data are depicted in Supplementary Movie 2, including standard deviations for each image.

**Proof-of-principle measurements of height and in-plane orientation with fluorescent beads.** As a first proof of concept and characterization of the method, we used fluorescence nanobeads immobilized in a mounting medium (see "Methods"), together with chosen polarizing elements prior to the SEO to simulate molecules with known orientations. The optical setup for CHIDO is displayed in Fig. 1b (see "Methods"). A 488-nm continuous laser is used for wide-field illumination of the sample via a high numerical aperture objective (NA = 1.45 oil immersion). The fluorescence ($\lambda = 520$ nm) is imaged onto an emCCD camera after passing through the SEO placed at the imaged backfocal plane of the objective. Importantly, nanobeads are also used to fine-tune the alignment of the SEO when used under circular polarization (see "Methods"). In such situation, we measured complementary rotationally symmetric PSF shapes in RHC and LHC channels (Fig. 5a), which are close to what is expected from theory (Fig. 5b).

Following alignment, measurements were taken for several sets of nanobeads (corresponding to different regions of the same sample) with polarization filters to simulate different fluorophore orientations. For each, measurements were taken at five defocus distances, at separations of 200 nm (see "Methods"). Rather than using a theoretical model, we chose the PSFs for one bead and used them to construct the PSF model used to extract the parameters for all the beads. The dependence in z of this PSF model was approximated by fitting the measured PSFs of the reference bead with a polynomial expression in z of the form

$$\mathcal{I}_n^{(p)}(\boldsymbol{\rho}, z) \approx \sum_{m=0}^{M} z^m \mathcal{I}_{n,m}^{(p)}(\boldsymbol{\rho}). \qquad (7)$$

For simplicity, we used an expansion up to $M = 2$, which is sufficient to fit the PSFs at five heights fairly well, but does introduce some systematic errors that limit the range in z over which the retrieval is valid. Details of this simple approximate approach for the estimation of z, its limitations, and ways to improve its range of validity are discussed in Supplementary

Note 4. Note that this method can be used as a starting point for a more rigorous parameter retrieval approach using the maximization of the likelihood function or the minimization of rms error. Finally, note that the retrieved direction parameters for polarized nanobeads do not include $\Omega$, since its value is expected to be close to 0, a region with known systematic bias (see Supplementary Figs. 6 and 7). Polarized nanobeads are a good framework to evaluate the robustness of the method for fixed dipoles.

We first investigated the case of emitters oriented in the axial z direction ($\theta = 0°$), whose polarization distribution at the pupil plane is radial. To simulate this situation, we inserted a radial polarization converter (Altechna, S waveplate) before the SEO. This experimental simulation could be made more accurate by also introducing an amplitude filter that simulates the correct radial dependence (approximately linear rather than constant). However, numerical simulations show that the difference in the resulting PSFs is not too significant. Images of four different sets of nanobeads were measured. A typical image taken at the central defocus position is shown in Fig. 5c. Using the references constructed from the PSFs of one bead from one of the sets, the transverse and axial positions of the nanobeads for all four sets were detected. Some of the results were discarded due to low confidence (calculated as the normalized correlation of the measured PSFs with those of the model evaluated at the estimated values of the parameter), caused by low signal levels, overlapping PSFs, or PSFs clipped at the edge of the field of view. The resulting number of nanobeads used for retrieval in set 1 was about 21 on average, while for the remaining sets, it was about 35. The average and standard deviations of the retrieved heights for each of the measurements are shown in Fig. 5d. For the four sets, the average estimated heights are separated by approximately 200 nm as expected. This result used a correction in which systematic errors were largely removed by replacing z with an appropriate monotonic function of z. (The results without the corrections are shown in Supplementary Fig. 9.) Note that from the retrieved 3D positions over the four sets, it was observed that the plane containing the nanobeads was tilted by about a quarter of a

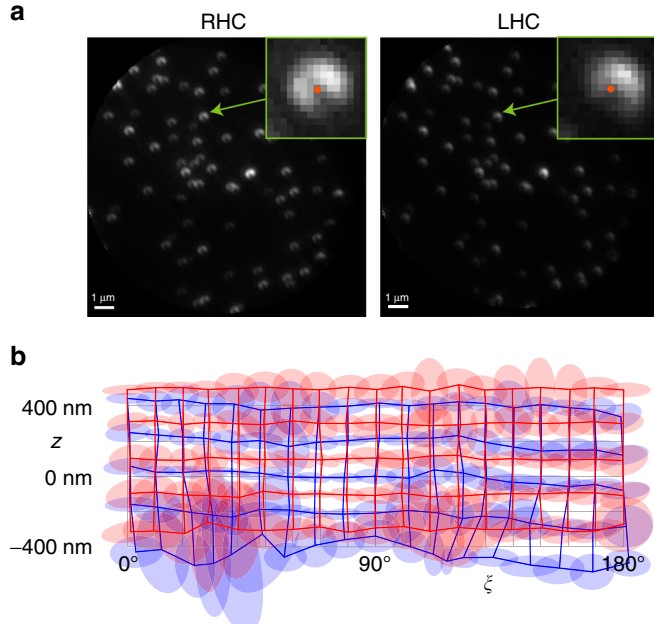

**b**

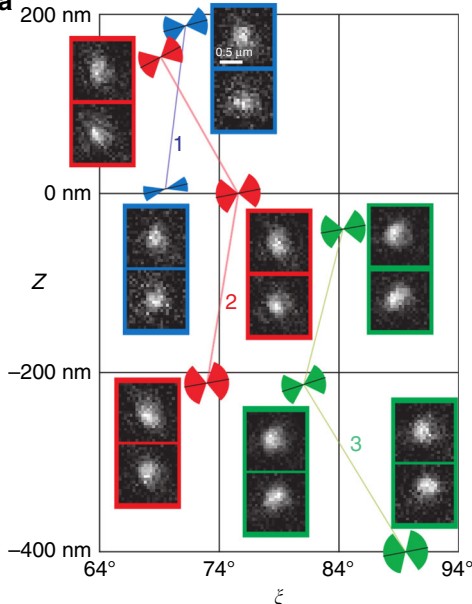

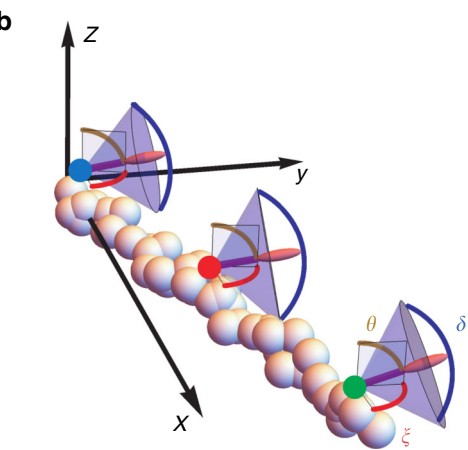

**Fig. 6 Experimental PSFs obtained from nanobeads under linear polarization for simulated oriented dipoles in the *xy* plane. a** Images for the two polarization components for set 2 at $\xi = 64°$ and at the central defocus position. The insets show zooms of the PSFs of a particular bead, where the red dot indicates the retrieved $(x, y)$ coordinates. (Note that the retrieved coordinates are not at the centroid of the measured PSFs.) For these images, the integration time was 1 s (camera gain 300). **b** The intersection points of the blue and red grids indicate the averages of the retrieved heights and orientation angles for each measurement, for sets 1 and 2, respectively, and the ellipses centered at each intersection indicate the corresponding standard deviations. A shift of 16° was applied to the $\xi$ axis so that the retrieved angles fall in the range [0°, 180°] for ease of interpretation (the polarization was rotated from −16° to 164°). The full set of data including standard deviations is shown in Supplementary Movie 3 for set 1 and Supplementary Movie 4 for set 2.

**Fig. 7 CHIDO imaging of three Alexa Fluor 488 single molecules sparsely attached to a F-actin filament. a** The insets show zooms of the images of the RHC/LHC components on the top/bottom for three different $z$ defocus (only high-confidence retrieval is shown). These images are labeled by the color of the frame: blue for molecule 1, red for molecule 2, and green for molecule 3. The estimated in-plane angle $\xi$ and height $z$ are represented as coordinates in this plot, where the PSFs used to retrieve the corresponding values are placed closest to the corresponding point. The out-of-plane angle $\theta$ and wobble angle $\delta$ are represented by the orientation and angular extent of the sketched cones. **b** 3D representation of the orientation and wobbling retrieval for defocus $z$ close to 0. Numerical values are given in Table 1.

degree. More details about the retrieved data from each set are shown in Supplementary Movie 2.

The number of photons detected for each nanobead was of the order of a hundred thousand, but with significant variations as can be appreciated from Fig. 5c and Supplementary Movie 2. Similarly, the SBR varied from about 1/2 to about 5. For each measurement, the spread in the estimates of $z$ was about 25–40 nm near $z = 0$ (depending on the set), growing to about 60 nm for $z \approx \pm 400$ nm (see Supplementary Fig. 10a). These spreads are due in part to the possible nonuniformity of the substrate's shape, or to variations in the size of nanobeads (nominally of 100 nm). The effect of some of these systematic sources of error can be removed by considering not the standard deviation of the estimated separations, but the differences of the height estimates at two consecutive heights for each bead, which brings the spreads down slightly, so that for some sets, the spread near $z = 0$ is of about 20 nm (Supplementary Fig. 10a). This is still significantly larger than the CRB predictions from the previous section, which for $\mathcal{N} = 50,000$ and SBR = 3 predict $\sigma_z \approx 3$ nm. Note, however, that part of the discrepancy emerges from the fact that the fluorescent beads have a size (100 nm) that is not negligible compared to the scale of the PSFs, and their extension both in the transverse and longitudinal directions has the effect of an appreciable blurring of the PSFs. By using the experimentally obtained PSF model and assuming $\mathcal{N} = 50,000$ and SBR = 3 we find instead $\sigma_z \approx 4.5$ nm

near $z = 0$ and 6–7 nm for $z = \pm 400$ nm, which is only off by about a factor of 4 from the measured standard deviations. This remaining factor is probably due to imperfections in the model, which was based on a single bead. For example, there could be small amounts of field-dependent aberrations over the measured field of view, which would deform the PSFs, therefore producing errors in the retrieval.

We then simulated emitters with different orientations within the *xy* plane (i.e., for $\theta = 90°$ and varying $\xi$) by replacing the S waveplate with a linear polarizer prior to the SEO (see "Methods"). Images were taken for two sets of nanobeads corresponding to two regions of a sample, each at 5 defocus

heights in steps of 200 nm, and for several orientations of the polarizer in steps of 10° over a range of 180°. One of these measurements is shown in Fig. 6a. Again, the measured PSFs from a single bead from one of the sets were selected to generate the PSF model used in the parameter retrieval for the others. Once more, a threshold in the level of confidence of the fit was applied to eliminate errors from overlapping/clipped PSFs and low signals, yielding results for about 30 nanobeads in set 1 and 36 in set 2. The insets in Fig. 6a show the retrieved $(x, y)$ position of a specific nanobead. The retrieved heights and orientations and their standard deviations for the two sets are shown in Fig. 6b, whose data are fully displayed in Supplementary Movie 3 and Supplementary Movie 4. An average defocus shift of about 100 nm was found between the two sets. We can also appreciate from the measurements that there was a relative drift in $z$ between both sets of about 100 nm over the time of data collection (over 30 min for each). Finally, it can be seen that the large standard deviations for some heights and directions in Fig. 6b are caused mostly by a few outliers not filtered out by the confidence threshold, corresponding to PSFs with low intensity, with overlaps, or clipped by the edge of the field of view. In general, we can see that the use of a quadratic approximation for $z$ gives rise to a magnification of the errors at the edges of the interval. The standard deviation in the estimate of $z$ varies greatly from image to image, with an average of about 39 nm. The corresponding standard deviation of the estimates of each bead's step sizes in $z$ is about 34 nm. The CR lower bound using the PSF model obtained from the beads is $\sigma_z \approx 7$ nm, which is about a factor of five smaller (see Supplementary Fig. 10b), the reasons for this factor being probably the same as those for the measurements using the S waveplate.

**Single molecules and super-resolution imaging**. We then applied CHIDO to super-resolution orientational imaging, using fluorophores appropriate for stochastic optical reconstruction microscopy (STORM)[38]. In order to evaluate the capacity of CHIDO to retrieve both 3D orientations and 3D positions of single molecules, we first imaged Alexa Fluor 488 fluorescent molecules (AF488) sparsely attached to in vitro-reconstructed F-actin single filaments via phalloidin (see "Methods"). These molecules are known to keep an average orientation along the actin filament, with a non-negligible wobbling extent[2].

For the first set of measurements, we retrieved the localization and orientation of the fluorophores by using a PSF basis set constructed from a combination of theoretical calculations and the reference nanobead measurements using the s-wave plate and linear polarizers. As explained in Supplementary Note 4, there are several potential problems with the resulting PSF model, which can arise from the combination of the models for in-plane and out-of-plane orientations, which used different beads whose relative 3D positions and brightnesses are not known accurately. Also, as mentioned before, beads have extensions that are much larger than that of single fluorophores, so the resulting PSF model is smoother. Figure 7 shows the results obtained for three molecules positioned along an F-actin filament. For each, three sets of images were taken at defocus separations of 200 nm. Isolated pairs of PSFs around the retrieved positions (after subtraction of the average background) are shown in the insets of Fig. 7a. The retrieved 3D positions, orientations, and wobble angles of these molecules are presented in Fig. 7a, b and in Table 1, with the exception of that for the top position for molecule 1, which fell outside the range of validity of the model generated from the nanobead reference measurements. The range of in-plane orientations $\xi$ measured for the three molecules is restricted to a 30° interval, which is expected from their

attachment to a single oriented filament. The off-plane angle $\theta$ and wobble angle $\delta = 2\arccos(1 - \Omega/2\pi)$ are also consistent with expectations: polarized measurements performed in 2D have shown fluctuations within an extent $\delta$ of about 90°, with a tilt angle with respect to the fiber that can reach 20°[2]. In the course of the measurements at different $z$ positions, the retrieved transverse positions (with respect to the center of the selected insets) present an uncertainty (namely the averaged standard deviation for each molecule, weighted by the number of measurements) of about 13 nm in both $x$ and $y$, and a corresponding directional uncertainty of about five degrees. The estimated defocus spacings average to 198 nm, but have an uncertainty of almost 50 nm. The uncertainty in the wobble solid angle is just below 0.9. Given the long integration times, a total of about 40,000 photons were detected for each molecule, with a SBR of about 1/3, so that according to the CR analysis, the uncertainties for the three spatial coordinates are about six times larger than the CR lower bound ($\sigma_{x,y} \approx 2.3$ nm and $\sigma_z \approx 7.5$ nm), and similarly for the directional parameters ($\sigma_\theta \approx 0.8°$ and $\sigma_\Omega \approx 0.13$). Recall that the PSF basis used for the retrieval was constructed from measurements for 100-nm beads, which do lead to slightly larger CR lower bounds than point dipoles.

Finally, we applied CHIDO to samples imaged in STORM conditions, e.g., single F-actin filaments labeled densely with AF488, within a buffer appropriate for on–off blinking conditions in order to localize individual emitters (see "Methods" and Supplementary Movie 5). The integration time was lowered to 200 ms, leading to significantly more challenging signal conditions than the single-molecule measurements described above. Estimates of 3D localization and orientation were performed on each detected single molecule by fitting the measured PSFs to theoretical model PSFs. We used a model based on theory in order to adapt to the slightly higher $c$ value used for this experiment (see "Methods"), and to overcome the limitations of reference PSFs generated with extended beads. Note that aberrations and misalignments in the system not accounted for by the model may induce some bias in the determined parameters, although not affecting precision. Figure 8a shows the rough positions of all detected molecules (color-coded by frame number) in a stack of about 5000 STORM image frames, on which single filaments are also identified by their low-resolution image. Note that after about 1000 frames, blinking seems to be dominated by molecules that are not attached to the filaments. In the collection of molecules measured, SBR values typically range from 0.2 to 1.2, with a large population around 0.3. Figure 8b depicts the resulting retrieved parameters for molecules within the highlighted section of the image, where we consider only molecules for which the normalized correlations of the measured PSFs (after background subtraction) to the model are above a threshold of 0.35. These molecules exhibit 3D and wobbling information that are in agreement with expected values. Notice that the range of retrieved heights is large, possibly due in part to inaccuracy introduced by the theoretical model used. Nevertheless, we can observe that the results are consistent with filaments laying on top of each other, as shown by the two molecules (C and E) near the junction of the two filaments, whose orientations are nearly perpendicular and whose heights are notably different. The measured PSFs for all these molecules are compared to those of the theoretical model evaluated at the retrieved parameters in Fig. 8c, which also shows the frame number, estimated number of photons, and normalized correlation between measured and model PSFs. The relatively large photon levels for some of the molecules depicted in Fig. 8c are due to the sum performed when a single molecule appears in several consecutive frames in the STORM image stack. Several molecules of Fig. 8c depict experimental PSFs that are slightly

**Table 1 Retrieved positions, orientations, and wobble angle for the fluorescent molecules in Fig. 7a, b.**

|  | Molecule 1 | | Molecule 2 | | | Molecule 3 | | | |
|---|---|---|---|---|---|---|---|---|---|
|  | $z_1$ | $z_2$ | $z_1$ | $z_2$ | $z_3$ | $z_1$ | $z_2$ | $z_3$ | St. dev. |
| $x$ (nm) | −108 | −98 | −72 | −93 | −109 | −132 | −118 | −114 | 12.3 |
| $y$ (nm) | 110 | 137 | 62 | 76 | 38 | 23 | 19 | 15 | 13.5 |
| $z$ (nm) | −2 | 221 | −218 | 8 | 121 | −499 | −268 | −75 | 49.6 |
| $\xi$ | 66° | 71° | 74° | 77° | 72° | 89° | 83° | 88° | 3.03° |
| $\theta$ | 77° | 75° | 80° | 79° | 62° | 79° | 72° | 80° | 5.78° |
| $\Omega$ | 0.4 | 1.5 | 3.2 | 2.3 | 1.7 | 3.7 | 1.8 | 2.2 | 0.88 |

The standard deviations on the last column are weighted averages of the standard deviations for each molecule, with the exception for that in z, which is the weighted average of the standard deviations for each molecule of the increments in height.

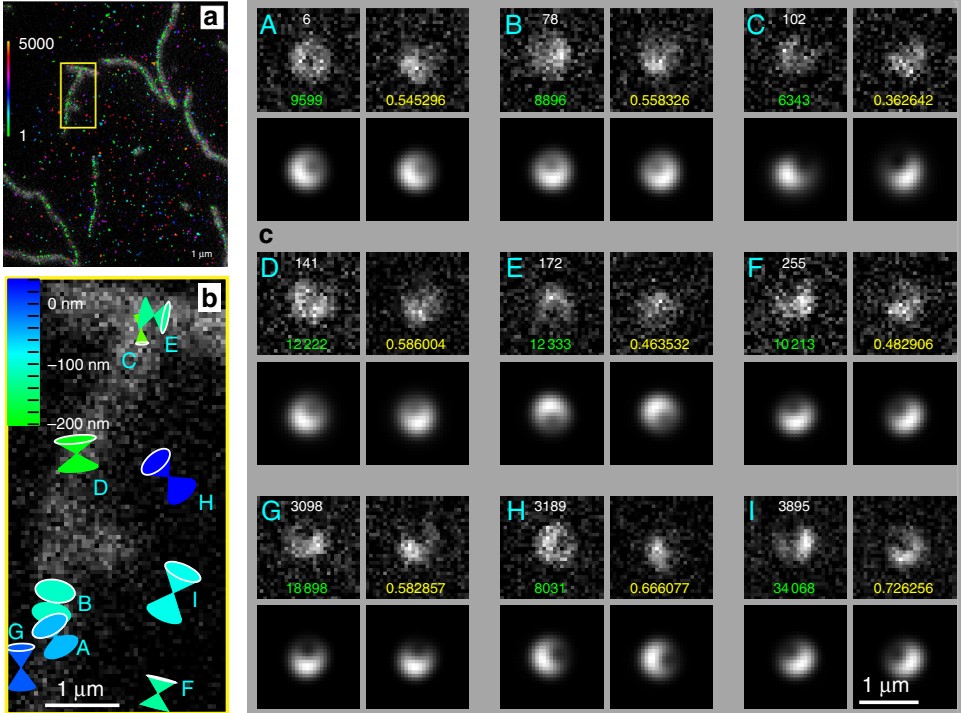

**Fig. 8 CHIDO analysis using STORM imaging on single F-actin filaments labeled with AF488 phalloidin conjugates. a** Detected single molecules in a STORM stack of about 5000 images, where the color indicates the stack number, shown over the low-resolution image of single filaments (gray background). **b** For the region within the yellow rectangle in (**a**), detected single molecules exhibiting a sufficiently high retrieval confidence level. The symbols correspond to 2D projection of 3D cone pairs whose vertex encodes x, y, whose color represents height according to the scale, whose orientation represents that of the fluorophore, whose solid angle represents Ω, and whose size reflects the correlation between the detected and model PSFs. **c** PSF pairs for each of the detected molecules in (**b**), where the top row shows the two detected PSFs and the bottom row the PSFs from the model evaluated at the retrieved parammeters. For each part, the white numbers indicate the frame at which the PSF appeared, the green number gives the estimated number of signal photons, and the yellow number is the correlation between the detected and model PSFs.

visually different than the retrieved PSFs, which we attribute to the presence of slight aberrations and/or misalignments (e.g., decentering of the SEO) in the optical setup, that are not included in the model. Such small imperfections can be the source of some degree of bias, that could be accounted for in a more complete model development that will be the focus of future efforts. Note that, according to the CR bounds, the highest precision of the retrieved positions is expected to be of the order of a nanometer and of the direction of about a degree. Furthermore, the precision in the solid angle wobble is of hundredths to about a tenth of a steradian. These expected levels of precision show that, when a proper reference model is used, CHIDO is applicable to STORM imaging conditions.

## Discussion

A method, CHIDO, was proposed that allows the measurement of the 3D position, averaged 3D orientation, and wobbling of isolated fluorophores, readily applicable for super-resolution orientational imaging. The key elements of this method are a specific spatially varying birefringent mask, the SEO, inserted at the pupil plane, and the subsequent separation of the two circular components to form separate images. The use of both images is shown to be of central importance for decoupling the estimations of in-plane orientation and axial position z. Despite the large amount of information encoded in the shape of the PSFs, these have dimensions that are only about twice as large as the corresponding diffraction-limited PSFs, making this approach suitable

for measurements with relatively high densities of molecules since their PSFs would not overlap significantly if their separations are about a micron or even less. The compactness of the PSFs also helps maintaining a higher SBR for the measured PSFs. Importantly, CHIDO is satisfactorily achromatic over the detected spectral range of fluorescence.

The retrieval of the parameters requires a reliable model for the PSFs. For the proof-of-concept measurements that emulate molecule orientation using beads and appropriate polarizers, the models were obtained by using the measurements from one of the beads and a polynomial interpolation in z. These references were also applied to the first set of measurements for single molecules. However, the construction of the PSF basis was based on reference measurements for orientations within the xy plane and normal to it, and a complete set of PSFs also requires measurements at intermediate off-plane angles (e.g., $\theta = 45°$), which are more difficult to simulate experimentally (one imperfect option being an off-center S waveplate). This incompleteness was addressed by using a mixture of theoretical simulations and experimental data. For the STORM-like measurements, however, a slightly higher value of c was used, which produced PSFs with finer features. We found that for such PSFs, the blurring resulting from using 100-nm reference beads was more critical, so instead we used a theoretically generated model. However, using a theoretical model that does not exactly account for specific characteristics of the imaging system might introduce systematic errors. In general, one of the main lines of research that we will explore in the future is the obtention of a reliable PSF model through a combination of theoretical and experimental approaches. These could include the use of smaller fluorescent beads combined with polarizers, appropriate deconvolution methods, and phase-retrieval techniques that use measurements at several heights. These models will also seek to characterize the effects of field-dependent aberrations, so that transverse position can be incorporated into the model beyond a simple (pixelized) translation.

A future direction to be explored is to use CHIDO not only to estimate the amount of wobbling of the molecules, but also the asymmetry of this wobbling[1,5]. As discussed in Supplementary Note 1, the $3 \times 3$ correlation matrix in Eq. (2) encodes information about the correlation of all field components[5], which in the context of vector coherence corresponds to the shape of an ellipsoid that characterizes the oscillations of the field[34,39]. Within the current context, this translates into the capacity of estimating not only a solid angle but, say, two angles of oscillation for the molecule supplemented by an angle of orientation of this elliptical cross section of the cone. We expect that with refinements of the system, and more importantly of the PSF basis, it will be possible to recover useful information about these extra parameters for single molecules, which can then be compared with computational models for the molecular motion. Finally, while CHIDO was restricted here to nonoverlapping PSFs, new fitting procedures could be developed to adapt the method to samples with higher densities, following recent work in the field[40–45].

With these possibilities, other applications for CHIDO can emerge in addition to imaging the 3D position and orientation of fluorophores. For example, this method could be used to probe the $3 \times 3$ correlation matrix at several points of a strongly nonparaxial field, such as a focused field or an evanescent wave. This would require the use of one or an array of subwavelength scatterers such as gold nanoparticles[46,47].

## Methods

**Optical setup**. The sample is excited by a laser (Coherent, Obis 488LS-20 for reference beads and single-molecule measurements; Coherent, Sapphire 488LP-200 for STORM measurements) in a wide field or TIRF illumination configuration

(Fig. 1b), and is held on a piezo nanopositioner (Mad City Labs Inc., Nano-Z200) to perform stacks along the z-axis with nanometric precision. Fluorescence light emitted by the sample is then collected by a ×60, NA 1.45 oil-immersion objective (Nikon, CFI Apo TIRF). A dichroic mirror (Semrock, DI02-R488) and a fluorescence filter (Semrock, 525/40) are used to select the emitted fluorescence and send it to the detection path. To adjust the field of view, a diaphragm is placed in a plane conjugate to the image. All the lenses are achromatic doublets: $L_1$ (125 mm) and $L_3$ (500 mm) are in a 4f configuration enabling us to locate the SEO in the back-focal plane; $L_2$ (400 mm) is used for back-focal plane imaging. To simulate emitters with different in-plane orientations, we put prior to the SEO a linear polarizer (Thorlabs, LPVISE100-A) mounted on a motorized rotation stage (Newport, PR50CC). To compensate unwanted polarization distortions introduced by the first dichroic mirror, we used another identical dichroic mirror (Semrock, DI02-R488), aligned along the plane where s and p polarization components of the incident fluorescence are inverted with respect to the incidence on the first dichroic mirror. Finally, the image is split into LHC and RHC polarization components by using a QWP followed by a quartz Wollaston polarizing 2.2° beamsplitter (Edmunds, 68-820), and each of these components is projected onto a different region of an emCCD camera (Andor iXon Ultra 897 for beads and single-molecule measurements; iXon Ultra 888 for STORM measurements). The total magnification provided by the lenses is 240, corresponding to a pixel size of 67 nm on the emCCD for the bead and single-molecule measurements, and 54 nm for the STORM measurements.

**Stress-engineered optic**. The term "stress engineering in optics" applies to the design and application of stress birefringence to achieve a desired retardance distribution. More specifically, the stress-engineered optic (SEO) used for this work was one of a collection of SEOs custom fabricated for the T.G. Brown research group at The Institute of Optics, University of Rochester. Details of the analysis and fabrication of the elements can best be found in the PhD dissertations of Alexis K. Spilman[48] and Amber M. Beckley[49] and summarized in several related publications[18,19,23]. To our knowledge, these elements are not yet commercially available, but can be readily manufactured by a skilled machine shop.

The window material can be any transparent material with a nonzero stress optic coefficient; we have used both fused silica and BK7 glass windows. The material must also be strong enough to withstand approximately 100 MPa of peripheral pressure without fracture. For the SEO used in this experiment, commercial BK7 windows (10-mm diameter, 8-mm thickness) were first given a fine grind in order to ensure a cylindrical edge. A metal ring (steel) with inner diameter of about 25 microns smaller than the outer diameter of the glass was cut on a lathe. An end mill was then used to remove material at 0°, 120°, and 240°, leaving three contact points at 60°, 180°, and 300°.

The assembly is accomplished by heating both the glass and metal to a temperature of 300 °C; at this temperature, the higher thermal expansion coefficient of the metal allows the insertion of the glass piece into the metal ring. Upon slow cooling, the ring then compresses the glass, applying force at three small regions around the perimeter in a way that is approximately uniform along the thickness.

**SEO alignment**. For the purpose of aligning the SEO and adjusting the parameter c, we used a sample of yellow highlighter's fluorescent ink, embedded in a mounting medium (Sigma, Fluoromount). The fluorescence emitted by this sample is used as a bright and homogeneous illumination for the SEO. Circular polarization was produced by placing a linear polarizer and QWP before the SEO. Also, a lens ($L_2$) was inserted to image the SEO plane, leading to complementary rotationally symmetric intensity patterns whose radial dependence for the two emerging circular components is approximately proportional to $\cos^2(cu/2)$ and $\sin^2(cu/2)$, respectively for $c = \pi$ (Fig. 1(c)). The system's alignment and calibration is then fine-tuned by removing $L_2$ and keeping the polarizer and QWP, to image model nanoemitters under circular polarization conditions. We used for this purpose fluorescent nanobeads of 100 nm in size (yellow-green carboxylate-modified FluoSpheres), immobilized on the surface of a poly-L-lysine-coated coverslip and covered with a mounting medium (Sigma, Fluoromount). Ideally, the resulting images are complementary, nearly rotationally symmetric PSF shapes, one of them donut-like, the other a bright spot, as shown in Fig. 5a, b[18]. These shapes are robust under defocus, but they are sensitive to polarization distortions, so they can also be useful for calibrating residual undesired birefringence. Once this stage of the calibration was complete, the polarizer and QWP prior to the SEO were removed.

**Single-molecule imaging**. To produce in vitro-reconstituted F-actin filaments, G-actin (AKL99, Cytoskeleton, Inc.) was polymerized at 5 μM in a polymerization buffer (5 mM Tris-HCl at pH 8.0, 50 mM KCl, 1 mM MgCl₂, 0.2 mM Na₂ATP at pH 7.0, and 1 mM DTT) in the presence of 5 μM phalloidin to stabilize the polymerization. To make the labeling sparse enough to isolate single molecules, we used a ratio of 1:200 phalloidin conjugated to Alexa Fluor 488. The filaments were then diluted to 0.2 μM, immobilized on the coverslip surface coated with poly-L-lysine, and covered with an imaging buffer containing an oxygen-scavenging system (5 mM Tris-HCl at pH 8.0, 50 mM KCl, 1 mM MgCl₂, 0.2 mM Na₂ATP at pH

7.0, 1 mM DTT, 1 mM Trolox, 2 mM PCA, and 0.1 μM PCD). The typical experimental conditions for single-molecule imaging were TIRF illumination, laser power of a few mW (at the objective plane), camera gain 300, and 1-s integration time.

**STORM imaging**. The F-actin filaments used for STORM imaging were obtained, as for the single-molecule images, from G-actin (AKL99, Cytoskeleton, Inc.) polymerized at 5 μM in a polymerization buffer (5 mM Tris-HCl at pH 8.0, 50 mM KCl, 1 mM MgCl$_2$, 0.2 mM Na$_2$ATP at pH 7.0, and 1 mM DTT). To fully label the actin monomers, the polimerization was done in the presence of 5 μM phalloidin conjugated to Alexa Fluor 488. The filaments were then diluted to 0.2 μM, immobilized on the coverslip surface coated with poly-L-lysine, and covered with a STORM buffer (100 mM Tris-HCl at pH 8.0, 10% glucose, 5 μ/ml pyranose oxidase, 400 μ/ml catalase, 50 mM β-MEA, 1 mM ascorbic acid, 1 mM methyl viologen, and 2 mM COT). Before taking images, the system was realigned, the value of $c$ was adjusted to 1.2$\pi$ (in order to benefit from slightly more complex PSFs), and the SEO was aligned so that one of its stress points pointed in the $y$ direction. The typical experimental conditions were TIRF illumination, laser power 150 mW, camera gain 300, and 200-ms integration time. For STORM imaging, a stack of 5000 images was used. For each frame, the approximate $x, y$ positions of the fluorophores were detected. Since some fluorophores blinked for longer than the exposure time of one image, a routine was written to sum over all the relevant consecutive frames for each fluorophore in order to reduce SNR. Some fluorophores blinked for up to about ten frames, resulting in photon numbers of up to about 50,000. Pairs of arrays of 29 × 29 pixels containing each PSF set were then used to retrieve the parameters.

## Data availability
All data are available from the corresponding authors upon reasonable request.

## Code availability
The code used is available from the corresponding authors upon reasonable request.

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

## Acknowledgements

This research has received funding from National Science Foundation (NSF) (PHY-1507278), Excellence Initiative of Aix-Marseille Université (AMU) A*MIDEX, a French "Investissements d'Avenir" programme, and European Union's Horizon 2020 research and innovation programme under the Marie Sklodowska-Curie grant agreement No 713750. Regional Council of Provence-Alpes-Côte d'Azur. A*MIDEX (No ANR-11-IDEX-0001-02) from the Investissements d'Avenir project funded by the French Government, managed by the French National Research Agency (ANR). CONACYT Doctoral Fellowship. The authors are grateful to P. Réfrégier who was instrumental in establishing this collaboration. They also thank A.J. Vella and M. Mavrakis, as well as A.M. Taddese, J. Puig, and M. Rahman for help in the method development. Additionally, the authors thank the Center for Integrated Research Computing (CIRC) at the University of Rochester for providing computational resources.

## Author contributions

M.A.A. and S.B. conceived and initiated the project, inspired on a polarimetry method by T.G.B. who designed and provided the SEO. S.B., L.A.C., and V.C. designed and built the optical system. V.C. and L.A.C. prepared the samples and performed experiments. M.A.A wrote the algorithm, performed the theoretical developments, and analyzed the data. L.A. C. and M.A.A. performed the Monte Carlo simulations. All authors wrote the paper and contributed to the scientific discussion and revision of the paper.

## Competing interests

The authors declare no competing interests.
