## [Peer Review File · Nature Communications]

Reviewers' comments:

Reviewer #1 (Remarks to the Author):

Single-molecule localization in optical microscopy lies at the heart of super-resolution methods such as PALM or STORM. In almost all applications of these techniques one assumes that the orientation of the labeling dyes is random or changes rapidly during their excited state lifetime, although a fixed dye orientation can severely bias the localization precision. One reason that most researchers assume a random dye orientation is that it is notoriously difficult to measure the three-dimensional orientation of single molecules. In the current manuscript, the authors introduce a new and sophisticated method for simultaneously determining the three-dimensional orientation AND localization of single emitters by using an unusual non-uniformly birefringent optical element. This element transforms the wavefront of collected fluorescent light in such a way that the image of a single emitter shows a distinct shape in the image plane that encodes for the orientation (and position) of the emitter. The authors explain in detail the theoretical principles behind their method, and present experimental results for imaging nanobeads (with appropriately tailored polarization properties) and three single molecules on an actin filament. An interesting technical detail of the presented method is that it can also estimate a potentially present restricted mobility (wobbling) of an emitter, which is difficult to do with other methods.

The method is very original and may find wider application in single molecule imaging and super-resolution microscopy, where simultaneous orientation and position measurements are important. One of the outstanding properties of the method is that, in the image of an emitter, the position and the orientation information are clearly decoupled, which theoretically ensures better accuracy in the determination of both. Thus, this new technique certainly deserves publication.

I have only two questions/comments that should be answered before publication:

1. The authors claim that their method is close to achromatic because the achromaticity of the imaging optics is compensated by that of the birefringent element. Would it be possible to present experimental measurements confirming this claim, for example using multicolor fluorescent beads?
2. The authors emphasize the potential of the method for three-dimensional single molecule imaging (and this is even part of the manuscript's title), but not much experimental data are shown for this claim. It would be great if the authors could present experimental data demonstrating the capability of their method for three-dimensional localization, for example by using labeled 2D-DNA origami samples.

Reviewer #2 (Remarks to the Author):

In this paper, Curcio, et al. propose a method to measure the 3D position, mean orientation, and wobble of fluorescent molecules. The authors used a stress-engineered optic (SEO) to modulate the electric field at the back focal plane of a microscope, followed by a Wollaston prism to separate right- and left-hand circularly polarized fluorescence. Termed Coordinate and Height super-resolution Imaging with Dithering and Orientation (CHIDO), the authors used polarization optics to modulate the fluorescence emission from beads in order to simulate dipole emitters and align/calibrate their imaging system. They next performed position and orientation measurements on a few fluorophores attached to F-actin filaments. Finally, the authors used a Monte-Carlo simulation to estimate the measurement precision of CHIDO.

I find the technical methodology to be quite interesting and believe the article falls within the scope of *Nature Communications*, but major revisions are needed to improve the work's technical content and presentation quality before it can be rigorously evaluated for publication. The primary concern is that there is insufficient simulation and experimental evidence for the claimed estimation performance advantages of CHIDO. Detailed recommendations to address this concern and others are below.

Major comments:

1. The authors need to revise their quantitative evaluation of the accuracy and precision of CHIDO using Monte Carlo simulations.
 - (a) The authors used an additive Gaussian noise model with different SNRs, which is an incorrect model for the photon counting process. The authors should reevaluate estimation performance using Poisson noise, where the mean and variance of the number of photons detected at each pixel j is given by a single parameter, e.g., λ_j . Please quantify the number of signal and background photons at each simulation condition instead of SNR since those numbers are more relevant to real experiments.
 - (b) Fig. 7 is difficult to read and interpret. Please quantify the statistical bias and precision

in measuring δ as a function of δ and comment on sources of bias.

- (c) No quantification is shown for the estimation performance of CHIDO on the polar orientation θ , the azimuthal orientation ξ , or the fluorophore 3D position (x, y, z) . Besides the precision in wobble angle estimation given in Fig. 7, the authors should quantify their estimator's accuracy and precision for all of these parameters at various signal and background levels. Comment on correlations and trends revealed in this analysis.
 - (d) The data in Supplementary Fig. 4 are unreadable. Please quantify measurement accuracy and precision separately for each estimation parameter, as above.
 - (e) Although the strength of CHIDO is the capability of measuring all the parameters simultaneously, it would be interesting to know how precise and accurate CHIDO is in estimating these parameters compared to some state of the art methods, e.g., those from Refs. 7-12. In particular, the authors claim optimality of their approach, e.g., in line 62, without any simulation or experimental evidence.
 - (f) The statistical study is based on Monte Carlo simulations (Fig. 7). However, the use of Fisher information and Cramér-Rao bound (CRB) are well-established in this field as an algorithm-agnostic approach to compare various methods. How well does CHIDO perform in terms of CRB compared to other orientation-measurement methods? How do the precisions of the authors' estimator in position and orientation compare to those predicted by CRB? If there is a discrepancy, please explain.
2. The decoupling of estimated parameters using CHIDO needs further justification. The authors claim that there is essentially no coupling in estimating transverse coordinates, axial coordinates, 3D orientation, and wobble angle of fluorescent probes. However, the PSF components and their gradients $\partial_{m,n}$ shown in Supplementary Fig. 1 are not orthogonal.
- (a) By "decoupled," do the authors mean that the parameters are statistically independent or just uncorrelated in the presence of measurement noise?
 - (b) In Supplementary Fig. 3, some estimation errors appear to be skewed ellipses, which means that they are correlated, as opposed to purely vertically or horizontally oriented ellipses. Even after cubic correction, there are still tilted ellipses in Fig. 5. Please explain.
 - (c) The cross correlation shown in Supplementary Fig. 2 implies that certain components

- are not orthogonal. This plot indicates that if we measure all 6 Stokes parameters together with the position, they are not uncorrelated. Please change the language in the main text to reflect this experimental non-ideality.
- (d) Further, please explore the (possible) correlations between the actual measurement parameters of interest: a 6×6 CRB/covariance matrix of x , y , z , θ , ξ , and δ for several reasonably chosen orientations and z positions. If the results show that the off-diagonal terms are much smaller than the diagonal terms, then perhaps the claim of uncorrelated measurements is justified.
 - (e) If the authors also want to claim that measurements of x , y , z , θ , ξ , and δ are orthogonal and independent of each other, then they should include additional panels in Fig. 5 showing σ_ξ (the precision in measuring ξ) vs. z and σ_z (the precision in measuring z) vs. ξ . If they are mostly flat, then these curves would imply that these parameters are independent.
 - (f) Please comment on the degraded/aberrated PSF images shown in Supplementary Fig. 1(c). How do these aberrations affect estimator performance? What is the likely source of these aberrations, are they position-dependent, and can they be easily corrected? These factors are critical in any experimental implementation of CHIDO, and should be discussed in the main text.
3. The authors used fluorescent beads combined with polarizers to mimic in-plane and out-of-plane molecules.
- (a) For in-plane molecules, the authors show the distribution of ξ measurements in Fig. 5. What about the out-of-plane angle θ and wobble δ estimates from this data set? Are these estimates consistent with immobile and in-plane dipoles? If not, please explain the discrepancy.
 - (b) The z standard deviations in Fig. 4 are very different across different groups for similar z positions. The standard deviation also varies quite a lot within the same group. Please provide z precision statistics of each group and comment on the trends. Please compare these experimental standard deviations to the theoretical localization precision, e.g., CRB. Are trends in these standard deviations correlated with the (x, y) position of each group? Please also comment on the degradation of localization precision for large Δz .

- (c) Please quantify the accuracy of z estimation in Figs. 4 and 5. The authors simply report that “the average estimated heights are separated by approximately 200 nm as expected.” Please precisely quantify these deviations. What is the likely source of deviations from expectation?
- (d) The z solid lines in Fig. 5 appear to be tilted. Is this tilt due to systematic bias as a function of ξ or simply system drift? Please improve the readability of this plot so that the absolute accuracy can be easily read.
- (e) For each experimental quantification of measurement precision, please state the average number of signal and photons detected.

4. The experimental demonstration of single molecules attached to F-actin filaments is weak.

- (a) The experimental demonstration contains only 8 orientation and z -position measurements of three molecules. Moreover, even though x , y , z , ϑ , ξ , and δ are measured, only ξ , δ , and z are shown. If CHIDO is to be used for single-molecule imaging, then the authors should demonstrate 3D position and orientation measurements of molecules attached to a continuous biological structure, e.g., similar to those in the authors’ previous work (Ref. 1).
- (b) The authors measured an in-plane orientation restricted to a 30 degree range for only 3 molecules. More molecules should be measured to justify that this measurement is consistent with other literature.
- (c) The authors should include the estimation precision and number of photons detected for all measurements shown in Table 1 for readers to understand the significance of the changes in the parameters measured.

Minor comments:

1. The polynomial expansion in Eq. 6 in the main paper only contains up to the second order terms. However, in the results in Fig. 5 use a cubic correction.

- (a) I think it would be clearer to write an infinite expansion here (similar to the equations in the SI), and state clearly the value of M (the order of the polynomial) used in each experiment.

- (b) The authors should also provide some insights regarding the conditions (e.g., z range) under which a quadratic approximation is sufficient, and when higher order terms are needed.
 - (c) The authors should include $m = 3$ elements in Supplementary Fig. 1 since they are used in the analysis.
2. Fig. 5 and Supplementary Fig. 3 are difficult to read. Is there a reason why blue and red are plotted in the same subplot? The authors could also scale the error ellipses to make them more readable.
 3. Please add scale bars to Figs. 2,3, 4ab, 5a and 6a.
 4. Please add color bars to Figs. 2,3, unless all colors in each panel use the same quantitative color scale. In that case, the authors should clearly state this fact in the caption.
 5. In the discussion section, the authors mention CHIDO's capability of measuring high densities of molecules, but this claim is not substantiated. They also postulate on its usefulness for multicolor imaging due to its approximate achromaticity. However, this use is also not shown. The authors should refine their language to state that these capabilities remain to be demonstrated in future work.
 6. The authors discuss possible measurements of asymmetric dipole wobbling in the discussion, but did not cite existing work in this area. The authors should cite Beausang, et al. *Biophys. J.*, 2013 and Ref. 10 here.
 7. What are the refractive indices of the experimental calibration sample (nanobeads) and the single molecule sample? A mismatch between them would almost certainly cause bias in the data analysis. Please comment.
 8. Eq. 2 in the SI mixes the unit vector \hat{x} and parentheses (x, y) representations of vectors. Please use consistent notations, e.g., $\cos(2\phi)\hat{x} + \sin(2\phi)\hat{y}$.
 9. The authors should cite seminal works on measuring single-molecule orientation, for example: J. E. Corrie, B. D. Brandmeier, R. E. Ferguson, D. R. Trentham, J. Kendrick-Jones, S. C. Hopkins, U. A. van der Heide, Y. E. Goldman, C. Sabido-David, R. E. Dale, S. Criddle, M. Irving, *Nature*. 1999.
H. Sosa, E. J. G. Peterman, W. E. Moerner, L. S. B. Goldstein, *Nat. Struct. Biol.* 2001 E. J. G. Peterman, H. Sosa, L. S. B. Goldstein, W. E. Moerner, *Biophys. J.* 2001

10. The authors should cite previous works when describing their forward imaging model, and state similarities and differences of their model compared to well-established models in the field (e.g., the relationship of the Stokes parameters to the second moments of dipole orientation):

Bohmer, Enderlein, J. Opt. Soc. Am. B, 2003

Lieb, Zavislan, Novotny, J. Opt. Soc. Am. B, 2004

Backer, Moerner, Opt. Express, 2015

Chandler, Shroff, Oldenbourg, La Rivière, J. Opt. Soc. Am. A, 2019

11. A high-density localization algorithm for single-molecule orientation imaging was not cited in line 291. Please add:

H. Mazidi, E. S. King, O. Zhang, A. Nehorai, M. D. Lew, 2019 IEEE 16th International Symposium on Biomedical Imaging (ISBI 2019)

12. Please correct several grammatical errors for clarity:

(a) Line 27: change “highly pixelated” to “finely sampled”

(b) Line 227: change “in in-plane” to “of in-plane”

(c) Line 316: change “alignement” to “alignment”

Reviewer #3 (Remarks to the Author):

This paper describes an elegant method to retrieve 3D position, orientation and wobbling by using an original phase mask. As orientation of fluorescent molecule can strongly bias its localization it's important to access this information without any coupling with spatial parameters. In my opinion the authors present an interesting solution, but some points should be modified. Detailed recommendations for improvement are below :

- Most of PSF engineering method suffers from field aberrations, is it the case for Chido ? Is there any evaluation on the localization precision obtained for the axial direction but also for the other parameters within the field of view ?

-Most of the experimental data (fig. 4 and 5) presented have been acquired on 100 nm beads with important acquisition time (1s) and gain (300), the authors should indicate the number of photons involves and how this approach could scaled down to PALM and dSTORM. Simulations in sup fig. 4 (which should be increased in size) indicates that the method could be applied to low number of photons, but some experimental results with a controlled and lower number of photons should be added.

-Supplementary Movie associated to figure 4 and 5 do facilitate the interpretation. Could the authors comment more quantitatively on the SNR threshold and density limit used to exclude some of the beads, as in the movie some of the point that seems to be excluded due to high density are taken into account for some of the positions, which could degrade the adjustment ? Would a global parameter on the whole stack be more appropriate to assess the density ?

-The method is declared to be robust to defocus, however on figure 5b, precision seems systematic lower at $z = -400$ nm ?

-From figure 5b , it's not easy to retrieve precise values on the axial precision, ellipses only give a rough idea, could a supplementary figure be added with the value of the axial precision represented precisely for various orientation ?

Fig 6 presents an application of isolated AF488 molecules, the insets zoom should be increased to truly appreciate the level of signal, and the number of photons should be indicated in the text.

Adding an image of a densely labeled sample (observed in PALM or dSTORM where the density can be controled to permit single moelcuel analysis) would be an important step to evidence the removed bias in a more complex structure, could the authors introduce this kind of sample in the current state of their development ?

Author's Detailed responses to reviewers

We would like to thank all three reviewers for their very thorough reading of our manuscript and their extremely useful comments. We believe that by addressing these comments the manuscript has improved very significantly. The two main changes are the replacement of the statistical analysis in the initial version with a study based on Cramér-Rao bounds, and the inclusion of new experimental results. We apologize for the time that these changes took, but in addition to having involved considerable modifications in the experimental setup, there were external factors that further delayed our capacity to resubmit the manuscript. We hope that the reviewers are satisfied with the result of these revisions. Below we provide detailed responses (in blue) to each comment by the reviewers.

Reviewer 1

Single-molecule localization in optical microscopy lies at the heart of super-resolution methods such as PALM or STORM. In almost all applications of these techniques one assumes that the orientation of the labeling dyes is random or changes rapidly during their excited state lifetime, although a fixed dye orientation can severely bias the localization precision. One reason that most researchers assume a random dye orientation is that notoriously difficult to measure the three dimensional orientation of single molecules. In the current manuscript, the authors introduce a new and sophisticated method for simultaneously determining the three dimensional orientation and localization of single emitters by using an unusual non-uniformly birefringent optical element. This element transforms the wavefront of collected fluorescent light in such a way that the image of a single emitter shows a distinct shape in the image plane that encodes for the orientation (and position) of the emitter. The authors explain in detail the theoretical principles behind their method, and present experimental results for imaging nanobeads (with appropriately tailored polarization properties) and three single molecules on an actin filament. An interesting technical detail of the presented method is that it can also estimate a potentially present restricted mobility (wobbling) of an emitter, which is difficult to do with other methods. The method is very original and may find wider application in single molecule imaging and super-resolution microscopy, where simultaneous orientation and position measurements are important. One of the outstanding properties of the method is that, in the image of an emitter, the position and the orientation information are clearly decoupled, which theoretically ensures better accuracy in the determination of both. Thus, this new technique certainly deserves publication.

We are delighted by the reviewer's very positive opinion of our work and by the fact that she/he appreciates clearly the main advantages of this method. We thank the reviewer for these comments and hope that the changes we made to the manuscript and Supplementary Materials, as well as our responses below, help address these comments.

I have only two questions/comments that should be answered before publication:

1. The authors claim that their method is close to achromatic because the achromaticity of the imaging optics is compensated by that of the birefringent element. Would it be possible to present experimental measurements confirming this claim, for example using multicolor fluorescent beads?

We believe it would be possible to present such measurements, and we plan to perform such experimental tests in the future. However, what we meant to emphasize is that the SEO modifies the wavefront via geometric phase effects caused by birefringence, and hence it does not require the use of a precise design wavelength. In other words, a small change in wavelength causes only a small change in PSF, so that using a broader spectrum does not affect greatly the encryption of orientation into PSF shape. To clarify further

this point, we now show in the Supplemental Materials simulations of the PSFs for a range of frequencies over the fluorescence spectrum. Even over this broad range of wavelengths the general shape of the PSFs remains fairly similar, and their extensions do not grow proportionally to the wavelength.

2. The authors emphasize the potential of the method for three-dimensional single molecule imaging (and this is even part of the manuscript's title), but not much experimental data are shown for this claim. It would be great if the authors could present experimental data demonstrating the capability of their method for three-dimensional localization, for example by using labeled 2D-DNA origami samples.

The initial submission did show measurements of single molecules over several different heights controlled experimentally, which is the most reliable way of validating our approach for 3D localization purposes. The longitudinal localization capabilities of the approach even allowed us to detect a sub-degree tilt in a sample plane containing fluorescent beads. Performing experimental assessment of 3D localizations requires well controlled structures, which is unfortunately delicate to obtain. DNA origami are for instance only partially controlled in their 3 dimensions (see for instance Deschamps et al. *Opt. Express* 22 (2014)) and labelled with fluorophores (usually complementary DNA strands conjugated with cyanines using DNA-PAINT) that are often not linked in a rigid way to the structure, making such experiments very delicate to control. We also believe that the addition in the revised version of calculations on the Cramer Rao bounds for both spatial and orientation estimations, as well as measurements on STORM imaging of many molecules attached to single F-actin filaments, ascertain the capacity of the method.

Reviewer 2

In this paper, Curcio, et al. propose a method to measure the 3D position, mean orientation, and wobble of fluorescent molecules. The authors used a stress-engineered optic (SEO) to modulate the electric field at the back focal plane of a microscope, followed by a Wollaston prism to separate right- and left-hand circularly polarized fluorescence. Termed Coordinate and Height super-resolution Imaging with Dithering and Orientation (CHIDO), the authors used polarization optics to modulate the fluorescence emission from beads in order to simulate dipole emitters and align/calibrate their imaging system. They next performed position and orientation measurements on a few fluorophores attached to F-actin filaments. Finally, the authors used a Monte-Carlo simulation to estimate the measurement precision of CHIDO.

I find the technical methodology to be quite interesting and believe the article falls within the scope of Nature Communications, but major revisions are needed to improve the work's technical content and presentation quality before it can be rigorously evaluated for publication. The primary concern is that there is insufficient simulation and experimental evidence for the claimed estimation performance advantages of CHIDO. Detailed recommendations to address this concern and others are below.

We are very happy that the reviewer finds the method interesting and appropriate for Nature Communications. He/she, however, indicates a series of points that should be clarified. We thank the reviewer for these comments, and we hope that the extensive revisions and new experimental results in the current version address the reviewer's concerns in a satisfactory way.

Major comments:

1. The authors need to revise their quantitative evaluation of the accuracy and precision of CHIDO using Monte Carlo simulations.
 - (a) The authors used an additive Gaussian noise model with different SNRs, which is an incorrect model for the photon counting process. The authors should reevaluate estimation performance using Poisson noise, where the mean and variance of the number of photons detected at each pixel j is given by a single parameter, e.g., A_j . Please quantify the number of signal and background photons at each simulation condition instead of SNR since those numbers are more relevant to real experiments.
 - (b) Fig. 7 is difficult to read and interpret. Please quantify the statistical bias and precision in measuring θ as a function of θ and comment on sources of bias.
 - (c) No quantification is shown for the estimation performance of CHIDO on the polar orientation θ , the azimuthal orientation ϕ , or the fluorophore 3D position (x, y, z) . Besides the precision in wobble angle estimation given in Fig. 7, the authors should quantify their estimator's accuracy and precision for all of these parameters at various signal and background levels. Comment on correlations and trends revealed in this analysis.
 - (d) The data in Supplementary Fig. 4 are unreadable. Please quantify measurement accuracy and precision separately for each estimation parameter, as above.
 - (e) Although the strength of CHIDO is the capability of measuring all the parameters simultaneously, it would be interesting to know how precise and accurate CHIDO is in estimating these parameters compared to some state of the art methods, e.g., those from Refs. 7-12. In particular, the authors claim optimality of their approach, e.g., in line 62, without any simulation or experimental evidence.
 - (f) The statistical study is based on Monte Carlo simulations (Fig. 7). However, the use of Fisher information and Cramér-Rao bound (CRB) are well-established in this field as an algorithm-agnostic approach to compare various methods. How well does CHIDO perform in terms of CRB compared to other orientation-measurement methods? How do the precisions of the authors' estimator in position and orientation compare to those predicted by CRB? If there is a discrepancy, please explain.

All of these points have been addressed by completely replacing the Monte Carlo simulations with an analysis based on Cramér-Rao bounds, as suggested by the reviewer, where the SNR is due exclusively to Poisson

noise. The results are indeed more interesting, informative and general, as they can be used to address the accuracy of all parameters for any number of photons. We thank the reviewer for suggesting these changes.

In particular, to address point (e) we compare the results of this CR analysis with the results in the cited references. We included a mention of this in the manuscript with the appropriate citations, but decided against describing in detail the comparison with each prior method within the manuscript. In this response, however, we can mention that, when estimating orientation parameters, CHIDO performs similarly to or better than other engineered-PSF methodologies without considerably increasing the PSF size. For example, for the *quadrated pupil* method (Backer, 2013) it is reported that for 920 photons they achieve $\sigma = 3.4^\circ$ and $\sigma = 2.4^\circ$; for a similar number of photons, both measures are about 2° for CHIDO, as shown in the new Fig. 4. For the *Trispot PSF* method (Zhang, 2018), for 3000 photons the reported values are $\sigma = 7^\circ$ and $\sigma = 8^\circ$. Techniques that do not use engineered phase masks, but are still based on PSF changes with defocus and orientation, like that by Aguet, 2009, have qualitatively similar CR bounds with respect parameter variations, although even with a small amount of defocus they worsen significantly. In the revised manuscript, we describe how the values vary within the 6-dimensional parameter space.

Please note that the claim of near-optimality of the SEO is not exactly for this method, but for prior applications in planar polarimetry, where by optimality we mean that the polarization information is fully encoded in the shape of the PSF *while causing a minimal increase to its extension*. After the initial submission of this manuscript, an article coauthored by one of the authors was published that provides this proof. This article is now cited as Ref. 24 in the revised version and the language regarding optimality was clarified. Given the steps in that proof, it is natural to see that the SEO is also nearly optimal for the current application, from the same point of view of encoding information while keeping the PSFs small.

2. The decoupling of estimated parameters using CHIDO needs further justification. The authors claim that there is essentially no coupling in estimating transverse coordinates, axial coordinates, 3D orientation, and wobble angle of fluorescent probes. However, the PSF components and their gradients $_{lm,m}$ shown in Supplementary Fig. 1 are not orthogonal.

The reviewer is correct in that perhaps the language we used was too strong, so we have modified it. However, please note from the top-left square in Supplementary Figure 2 that, for $z = 0$, the PSFs are indeed largely uncoupled, since the matrix is essentially diagonal except for some coupling between $m = 0$ and $m = 8$. Please note, also, that the 6 elements shown are more than the parameters they encode (the three angular parameters), so that coupling of some of the PSFs do not imply coupling of the parameters. Nevertheless, the level of coupling of the relevant parameters is best characterized by the CRB analysis, which has now been added to the manuscript. There is indeed some small amount of coupling between some of the parameters for specific values, but these are never significant, as discussed in the revised version.

- (a) By "decoupled," do the authors mean that the parameters are statistically independent or just uncorrelated in the presence of measurement noise?

We meant statistically independent. We think the inclusion of the CRB analysis helps clarify this.

- (b) In Supplementary Fig. 3, some estimation errors appear to be skewed ellipses, which means that they are correlated, as opposed to purely vertically or horizontally oriented ellipses. Even after cubic correction, there are still tilted ellipses in Fig. 5. Please explain.

There is skewness introduced by the polynomial expansion in z , particularly towards the edges. We would like to stress, however, that some of the skewness in this figure is not inherent to the optical implementation but results from the particular simplified algorithm used here for the retrieval in z . The fundamental levels of skewness for the proposed optical technique are those characterized by the CR bounds.

- (c) The cross correlation shown in Supplementary Fig. 2 implies that certain components are not orthogonal. This plot indicates that if we measure all 6 Stokes parameters together with the position, they are not uncorrelated. Please change the language in the main text to reflect this experimental non-ideality. We have modified the language to avoid any misunderstanding. Nevertheless, please notice that the fact that the 18 PSFs in this figure are not orthogonal does not necessarily imply a coupling of the

parameters they encode, which are only 4 (the three angular parameters and z). Further, the largest coupling is between the PSFs for $m = 0$ and $m = 2$, and this was to be expected, because the $m = 2$ PSFs are somewhat redundant (in the sense that only two sets of PSFs are strictly required to estimate the parameter z) and were added to avoid negativity in the PSF model for large values of z . Let us stress again, however, that the true level of coupling is that found through the CRB analysis.

- (d) Further, please explore the (possible) correlations between the actual measurement parameters of interest: a 6×6 CRB/covariance matrix of x , y , z , θ , ϕ , and δ for several reasonably chosen orientations and z positions. If the results show that the off-diagonal terms are much smaller than the diagonal terms, then perhaps the claim of uncorrelated measurements is justified.

We thank the reviewer for this suggestion, which has now been implemented. As shown in the new section, the magnitude of the off-diagonal elements is on average less than 10% of the diagonal ones, and at their peak at localized parameter values they are at most about 25% for selected pairs of parameters.

- (e) If the authors also want to claim that measurements of x , y , z , θ , ϕ , and δ are orthogonal and independent of each other, then they should include additional panels in Fig. 5 showing σ_x (the precision in measuring x) vs. z and σ_z (the precision in measuring z) vs. x . If they are mostly flat, then these curves would imply that these parameters are independent.

This type of plot is now included in the section devoted to the CRB analysis. Please note that the uncertainties in the old Fig. 5 are not the most appropriate place for this analysis, because those uncertainties are affected by a series of other factors, including not only the current limitations of the retrieval technique mentioned earlier, but also the fact that the ellipses in the figure are calculated from collections of beads, whose actual positions are not well known. The plots in the new section on CRB are, however, more indicative of the fundamental limitations of the proposed method. Both the experimental data and the CRB analysis indicate that while indeed independence is not complete, the level of coupling is quite small and causes little compromise in the estimation of the parameters.

- (f) Please comment on the degraded/aberrated PSF images shown in Supplementary Fig. 1(c). How do these aberrations affect estimator performance? What is the likely source of these aberrations, are they position-dependent, and can they be easily corrected? These factors are critical in any experimental implementation of CHIDO, and should be discussed in the main text.

The distortions come partly from aberrations, but also from the fact that it is difficult to estimate coupling between in-plane and out-of-plane components. Also, the polarizers (particularly the s -plate) used to obtain the references do not perfectly reproduce the desired pupil distribution. Their main effect is to cause systematic errors in the determination of θ . Please note, however, that the way we chose to show the figure in the initial submission, in which each PSF is normalized, exacerbated the distortion. For example, for $m = 1$ the most severely distorted PSFs are those for $n = 0$ and 4. However, these play a minor role in the retrieval process because their range of values is much smaller than for the rest. In the corresponding figure in the revised version, each row is now normalized to the maximum value for that m . The PSFs for $m = 2$ also show more visible distortions, but these are also less critical for the determination of the parameters, although they are partly responsible for the range of errors seen in the bead measurements and the need for a subsequent correction for systematic errors. As the reviewer requests, we now discuss in the main manuscript the possible adverse effects of problems with the PSF model. These, however, are not fundamental to the method but are specific to a particular implementation and will be improved with better methods to calculate the basis (as detailed in the Discussion section), which will be one of our main objectives in future research.

3. The authors used fluorescent beads combined with polarizers to mimic in-plane and out-of-plane molecules.

- (a) For in-plane molecules, the authors show the distribution of measurements in Fig. 5. What about the out-of-plane angle θ and wobble δ estimates from this data set? Are these estimates consistent with immobile and in-plane dipoles? If not, please explain the discrepancy.

We chose to present initially the consistency of these measurements separately, as they are building blocks for the experimentally-motivated PSF set. Because the shapes of the in-plane and out-of plane PSFs are so different and the PSFs are so uniform over the field of view, the best fits correspond to in-plane and out-of-plane dipoles, so the coupling and the resulting error introduced in θ and wobble are not too significant (a few degrees and fairly systematic, particularly in θ ; such systematic error was used when interpreting single molecule data). However, we are aware that this fact is not proof that the PSF model is good for estimating θ (and wobble) for intermediate orientations, but only that the PSFs for purely in-plane and purely out-of-plane are clearly linearly independent (even if not exactly orthogonal). The main challenge for future work is the acquisition of a reliable PSF basis, particularly of the elements responsible for the coupling of in-plane and out-of-plane. This is now stressed further in the revised version.

- (b) The z standard deviations in Fig. 4 are very different across different groups for similar z positions. The standard deviation also varies quite a lot within the same group. Please provide z precision statistics of each group and comment on the trends. Please compare these experimental standard deviations to the theoretical localization precision, e.g., CRB. Are trends in these standard deviations correlated with the (x, y) position of each group? Please also comment on the degradation of localization precision for large $|z|$.

These variations are partly caused by problems with the experimentally estimated PSFs, particularly the polynomial expansion, and the limited range of z used for calibration, which introduced errors. These errors are more pronounced away from the central z , the bottom z measurement for set 1 having an anomalously large spread. However, particularly near the nominal plane, the observed standard deviation might not be due entirely to imprecision in the retrieval, but also to the fact that the vertical position of each of the beads is not known to the required precision; the substrate to which they are attached might not be perfectly flat, the beads themselves might not be perfectly round or all of exactly of the same size (their nominal standard deviation being 5%). Please note that the diameter of the beads is much larger than the standard deviations we are seeing near the nominal $z = 0$ plane. As the reviewer suggests, there is correlation with the (x, y) coordinates (for example, the beads at the bottom of the image for set 1 seem to consistently be below the best-fit plane) as z is varied, which suggests that part of the global standard deviations is in fact due to the bead centroids not being exactly coplanar. Another factor might be the precision of the translation piezo-stage, which is also of the order of nanometers. The standard deviations we see near the nominal plane, which can be of about 25-30 nm, are therefore not too surprising. We added in the caption and description the range of values, and indicated that the specific values for each orientation and height are specified in the Supplementary Movies 2, 3, and 4. We now also comment in the revised version how these standard deviations of ensembles of beads compare with the uncertainties from the CR analysis, the latter being more indicative of the fundamental accuracy of CHIDO.

- (c) Please quantify the accuracy of z estimation in Figs. 4 and 5. The authors simply report that "the average estimated heights are separated by approximately 200 nm as expected." Please precisely quantify these deviations. What is the likely source of deviations from expectation?

The standard deviations are implicit in the error ellipses, but we added a mention to specific numbers. The possible causes for these standard deviations are essentially the same as those described in the previous point: a combination of deficiencies in the model used for the retrieval, particularly the quadratic model for the dependence in z , and variations in the actual positions and even sizes and shapes of the beads. An extra possible source of error for these measurements comes from the rotation of the polarizer. We now provide numbers for the representative standard deviations and indicate that more details of these deviations are given in the Supplementary Movies.

- (d) The z solid lines in Fig. 5 appear to be tilted. Is this tilt due to systematic bias as a function of ξ or simply system drift? Please improve the readability of this plot so that the absolute accuracy can be easily read.

We calibrated with respect to the red set, to make it self-consistent. The tilted lines were probably caused by a different drift in z for each of the two sets. We expect that our setup is susceptible to this

type of drift over time (over 30 minutes between the initial and final directions for each set). This is now discussed in the manuscript.

- (e) For each experimental quantification of measurement precision, please state the average number of signal and photons detected.

This has been added.

4. The experimental demonstration of single molecules attached to F-actin filaments is weak.

The goal of this section was initially to provide a proof of principle. To emphasize the possible use of CHIDO for super-resolution methods based on single molecules localization, we supplemented these early measurements with STORM-like measurements. However, we decided to retain the single molecule measurements since they confirm the ability to estimate z with consistency for the other parameters.

- (a) The experimental demonstration contains only 8 orientation and z -position measurements of three molecules. Moreover, even though x , y , z , θ , ϕ , and δ are measured, only ϕ , δ , and z are shown. If CHIDO is to be used for single-molecule imaging, then the authors should demonstrate 3D position and orientation measurements of molecules attached to a continuous biological structure, e.g., similar to those in the authors' previous work (Ref. 1).

All parameters are now represented and specified in detail in Table 1, including the displacements in z . Please notice that θ is also shown in the figure as inclination; there was an error in the caption that indicated that this inclination corresponded to ϕ .

- (b) The authors measured an in-plane orientation restricted to a 30° range for only 3 molecules. More molecules should be measured to justify that this measurement is consistent with other literature.

We have added a new set of STORM measurements, which include many molecules.

- (c) The authors should include the estimation precision and number of photons detected for all measurements shown in Table 1 for readers to understand the significance of the changes in the parameters measured.

The average number of photons detected is now indicated. All molecules had similar numbers of photon counts.

Minor comments:

1. The polynomial expansion in Eq. 6 in the main paper only contains up to the second order terms. However, in the results in Fig. 5 use a cubic correction.

This is true, but the PSFs used corresponded to the quadratic expression. A mapping of the measurements according to reference measurements required the inclusion of a cubic term. We have modified the description to clarify that the cubic in question does not correspond to an extra term in Eq. (6).

- (a) I think it would be clearer to write an infinite expansion here (similar to the equations in the SI), and state clearly the value of M (the order of the polynomial) used in each experiment.

We followed the reviewer's comment, but rather than an infinite series, we left the upper value as M .

- (b) The authors should also provide some insights regarding the conditions (e.g., z range) under which a quadratic approximation sufficient, and when higher order terms are needed.

We now discuss this in more detail. The essential point is how well the expansion matches the known PSFs. $M = 2$ is only valid for relatively small ranges (of the order of a fraction of a Rayleigh range) around the nominal plane, since then the PSFs begin to expand. We are now exploring other sets of functions for the interpolation as described briefly in the Supplementary Materials, but we believe the polynomials are sufficient for the proof of principle in this manuscript, and convenient for presenting the main ideas.

(c) The authors should include $m = 3$ elements in Supplementary Fig. 1 since they are used in the analysis. Let us stress that $m = 3$ was not used in the analysis. The mapping means that, in the polynomial expansion, z is replaced by a simple, monotonic function of z , and this function includes a cubic term. However, the only PSFs that were used were $m = 0, 1, 2$. We tried to clarify the language in the manuscript to avoid confusion.

2. Fig. 5 and Supplementary Fig. 3 are difficult to read. Is there a reason why blue and red are plotted in the same subplot? The authors could also scale the error ellipses to make them more readable.

We thought this was a good idea to show the consistency of both sets. The error ellipses are in the same units as the scale, so we think scaling them would be misleading.

3. Please add scale bars to Figs. 2,3, 4ab, 5a and 6a.

This has been done.

4. Please add color bars to Figs. 2,3, unless all colors in each panel use the same quantitative color scale. In that case, the authors should clearly state this fact in the caption.

This has been done.

5. In the discussion section, the authors mention CHIDO's capability of measuring high densities of molecules, but this claim is not substantiated. They also postulate on its usefulness for multicolor imaging due to its approximate achromaticity. However, this use is also not shown. The authors should refine their language to state that these capabilities remain to be demonstrated in future work.

The claim of high density was not substantiated with experiments in the original submission, but we felt that it was not necessary as the claim was based on the compactness of the PSFs, which is indeed shown theoretically and experimentally. We have qualified the statement to clarify this point.

The comment on achromaticity was added not so much because of the possibility for multicolor imaging but for justifying using monochromatic PSF models instead of ones accounting for the used fluorescence spectrum. We have added a figure to the Supplementary Materials to illustrate this fact. There was a passing comment on the possible use of this property in the concluding remarks. We have modified this comment to clarify that this will be explored in future work.

6. The authors discuss possible measurements of asymmetric dipole wobbling in the discussion, but did not cite existing work in this area. The authors should cite Beausang, et al. Biophys. J., 2013 and Ref. 10 here.

We have added references to both this article and to the 2015 article by Backer et al. in the relevant discussions on asymmetric dipole wobbling. We thank the reviewer for this suggestion.

7. What are the refractive indices of the experimental calibration sample (nanobeads) and the single molecule sample? A mismatch between them would almost certainly cause bias in the data analysis. Please comment.

Distances that were changed by moving the z translation stage imply changes of the thickness in the immersion oil between the sample and the microscope, while differences in z between different molecules correspond to distances in a water-based solution. The effect of these refractive indices as well as the corresponding distributions of plane wave directions were taken into account in the models.

8. Eq. 2 in the SI mixes the unit vector \hat{x} and parentheses (x, y) representations of vectors. Please use consistent notations, e.g., $\cos(2\phi)\hat{x} + \sin(2\phi)\hat{y}$.

This has been corrected.

9. The authors should cite seminal works on measuring single-molecule orientation, for example:

J. E. Corrie, B. D. Brandmeier, R. E. Ferguson, D. R. Trentham, J. Kendrick-Jones, S. C. Hopkins, U. A. van der Heide, Y. E. Goldman, C. Sabido-David, R. E. Dale, S. Criddle, M. Irving, Nature. 1999.

H. Sosa, E. J. G. Peterman, W. E. Moerner, L. S. B. Goldstein, Nat. Struct. Biol. 2001

E. J. G. Peterman, H. Sosa, L. S. B. Goldstein, W. E. Moerner, Biophys. J. 2001

We have included these references. We thank the reviewer for pointing them out.

10. The authors should cite previous works when describing their forward imaging model, and state similarities and differences of their model compared to well-established models in the field (e.g., the relationship of the Stokes parameters to the second moments of dipole orientation):

Bohmer, Enderlein, J. Opt. Soc. Am. B, 2003

Lieb, Zavislan, Novotny, J. Opt. Soc. Am. B, 2004

Backer, Moerner, Opt. Express, 2015

Chandler, Shroff, Oldenbourg, La Rivière, J. Opt. Soc. Am. A, 2019

We have added these references too. The relationship between the Stokes parameters and the second moments is direct, as is appreciable from Eq. (2) which is precisely the matrix of second moments. To stress this relation, we now use the words "second moments" when we introduce the correlation matrix.

11. A high-density localization algorithm for single-molecule orientation imaging was not cited in line 291. Please add:

H. Mazidi, E. S. King, O. Zhang, A. Nehorai, M. D. Lew, 2019 IEEE 16th International Symposium on Biomedical Imaging (ISBI 2019)

This reference has been added.

12. Please correct several grammatical errors for clarity:

(a) Line 27: change "highly pixelated" to "finely sampled"

(b) Line 227: change "in in-plane" to "of in-plane"

(c) Line 316: change "alignement" to "alignment"

We thank the reviewer for catching these errors and inaccuracies.

Reviewer 3

This paper describes an elegant method to retrieve 3D position, orientation and wobbling by using an original phase mask. As orientation of fluorescent molecule can strongly bias its localization it's important to access this information without any coupling with spatial parameters. In my opinion the authors present an interesting solution, but some points should be modified. Detailed recommendations for improvement are below:

We thank the reviewer for the useful comments.

1. Most of PSF engineering method suffers from field aberrations, is it the case for CHIDO? Is there any evaluation on the localization precision obtained for the axial direction but also for the other parameters within the field of view?

This is an interesting comment. We think that due of the smoothness of the SEO mask (which is also important for the compactness of the PSFs) these field-dependent aberrations are probably not as important as for other more complex phase masks. There is evidence that field-dependent aberrations are not affecting too significantly the PSFs from our measurements using beads with polarizers, which look very similar at the edges with respect to the center. However, we cannot rule out that some of the standard deviations we observe in Figs. 5 and 6 might be partly caused by these aberrations. If this were the case, these could be well characterized and the PSF model could include corrections that depend on x , y . This is a very good suggestion that we will take into consideration in our future work on refining the reference PSFs.

We think that our setup might have other (field-independent) aberrations, such as a small amount of spherical aberration, which breaks slightly the symmetry around the nominal $z = 0$ plane of the PSF shapes. However, as is now discussed in the Cramér-Rao bound analysis, a small amount of this type of aberration does not affect significantly the accuracy of the measurements as long as it is incorporated in the reference PSFs.

2. Most of the experimental data (Fig. 4 and 5) presented have been acquired on 100 nm beads with important acquisition time (1s) and gain (300), the authors should indicate the number of photons involves and how this approach could scaled down to PALM and dSTORM. Simulations in sup Fig. 4 (which should be increased in size) indicates that the method could be applied to low number of photons, but some experimental results with a controlled and lower number of photons should be added.

These measurements were performed under relatively low power conditions. We have now added STORM-like measurements (under proper laser power conditions), as a proof of principle that this technique can work with the appropriate numbers of photons. The old Fig. 4 in the Supplementary Materials has been removed, because the statistical analysis was replaced with the CRB analysis in the main manuscript.

3. Supplementary Movie associated to Figures 4 and 5 do facilitate the interpretation. Could the authors comment more quantitatively on the SNR threshold and density limit used to exclude some of the beads, as in the movie some of the point that seems to be excluded due to high density are taken into account for some of the positions, which could degrade the adjustment? Would a global parameter on the whole stack be more appropriate to assess the density?

What was used as a metric to exclude some beads was the correlation of the fit of the model PSFs to the measured PSFs. This way, if there are overlapping PSFs, PSFs clipped at the image's edge, or if the photon number is too low, the agreement of the fit and the measurement is not good and the bead is discarded. This is a simple measure but it is not perfect: sometimes it discards fairly good PSFs, while it retains overlapping PSFs. These problems surely affected the error estimates in the figures. We now comment on this fact. However, we did not put too much effort yet into refining the algorithm because we regard the optical technique as the main contribution of this manuscript. Future work will focus on such practical improvements to the algorithms.

Regarding density, the only requirement is that the PSFs do not overlap, so it is quite easy to see from the images which PSFs would be problematic. For the NA and values of c being used in this work, the minimal distance is of the order of 500 nm to a micron, as we now mention in the concluding remarks. In

this respect, we expect that by using appropriate, more elaborate algorithms it would be possible to retrieve data from PSFs with some level of overlap, but this is beyond the scope of this manuscript.

4. The method is declared to be robust to defocus, however on Figure 5b, precision seems systematic lower at $z = \pm 400$ nm?

There is an increase in uncertainty for spatial localization as we move away from the nominal $z = 0$ plane, as shown by the CR bound included in the revised version. However, the effect observed in the old Fig. 5b (now Fig. 6b) is due to the quadratic model in z losing validity near the edge of the interval. This is now discussed in more detail in the manuscript.

5. From Figure 5b, it's not easy to retrieve precise values on the axial precision, ellipses only give a rough idea, could a supplementary figure be added with the value of the axial precision represented precisely for various orientation?

More precise data is provided in the Supplementary movies 2, 3 and 4, for each image, both before and after subtracting the spread due to the tilt of the plane. The precision varies from under 30 nm over 80 nm, depending on the height and orientation. As discussed now in the manuscript, these errors are well above the CR bounds and can surely be reduced by an improvement in the reference PSFs. However, as we described in our response to Reviewer 2, the spreads in the old Fig. 5b are not strictly speaking precisions, but standard deviations of estimates of the positions of ensembles of beads, whose size is larger than the standard deviations and whose exact z positions are not known and not guaranteed to be the same.

6. Figure 6 presents an application of isolated AF488 molecules, the insets zoom should be increased to truly appreciate the level of signal, and the number of photons should be indicated in the text.

We now show many measured PSFs for STORM measurements of single molecules to better address this point. They are shown in the new Fig. 8, and in the Supplementary Movies 6-8. The photon numbers are also indicated.

7. Adding an image of a densely labeled sample (observed in PALM or dSTORM where the density can be controlled to permit single molecule analysis) would be an important step to evidence the removed bias in a more complex structure, could the authors introduce this kind of sample in the current state of their development?

We added this type of measurement in the revised version. These new measurements show that the method can produce very clear PSFs encoding the location and orientation of multiple molecules.

Reviewer #1 (Remarks to the Author):

The authors have adequately answered all questions of the referees. The additional changes and modifications improved the manuscript significantly. I recommend publication as is.

Reviewer #2 (Remarks to the Author):

In this paper, Curcio, et al. propose a method to measure the 3D position, mean orientation, and wobble of fluorescent molecules. The authors used a stress-engineered optic (SEO) to modulate the electric field at the back focal plane of a microscope, followed by a Wollaston prism to separate right- and left-hand circularly polarized fluorescence. Termed Coordinate and Height super-resolution Imaging with Dithering and Orientation (CHIDO), the authors used Cramér-Rao bound calculations to predict the precisions of localizing single molecules in 3D, as well as measuring their orientation and wobble. They use polarization optics to modulate the fluorescence emission from beads in order to simulate dipole emitters and align/calibrate their imaging system. Finally, CHIDO is experimentally demonstrated by measuring the position and orientation of a collection of fluorophores attached to F-actin filaments.

The revised manuscript is significantly improved over the original version – the Cramér-Rao bound calculations show that the concept can theoretically achieve localization and orientation measurement precisions on par with existing techniques. However, my major concern lies with the experimental robustness of CHIDO. There is not enough evidence to conclude that the method can perform well, especially in terms of the accuracy and precision of measuring fluorophore z position and wobbling angle, in single-molecule super-resolution imaging. Measuring 3D position and wobble simultaneously are the principal innovations of the proposed approach, but they are not demonstrated convincingly. Detailed recommendations to address this concern and others are below.

Major comments

1. Repeatedly throughout the manuscript, the authors write “accuracy” inappropriately when they should use the term “precision.” Examples include lines 206, 208, 210, 212, 332, 333, and 363. Measurement accuracy is the bias or error in the measurement when noise is absent or infinitesimal. Statistical precision is the variation or standard deviation of a set of repeated measurements taken under identical conditions. These two errors are separate but are both important to quantify individually. Moreover, Cramér-Rao bound (CRB) analysis assumes that an unbiased (accurate) estimator is used, and thus predicts the best-possible variance (precision) of such an estimator. The CRB cannot be used to prove that a technique is accurate. Further, one must show that his/her estimator is unbiased to infer that any CRB analysis applies to it. Please revise the manuscript to distinguish between these concepts appropriately.
2. More theoretical and experimental evidence is needed to show that molecular wobbling/dithering can be measured accurately and precisely.
 - a. The authors report the Cramér-Rao bound (CRB) crp3D of measuring a generalized 3D Stokes parameter (Fig. 4) instead of directly characterizing of the uncertainty of measuring the cone solid angle θ (reported in Fig. 8) or wobble angle S (reported in Fig.

- 7). This disagreement between theoretical predicted performance and actual experimental measurements makes it difficult to evaluate how well the technique performs in actual imaging experiments.
- i. Please calculate the CRB for measuring θ and report these values in Fig. 4.
 - ii. For the signal and background level of the experiments in Figs. 7 and 8, what is the CRB-predicted precision of estimating θ ?
 - iii. How well does the estimator used on the data in Figs. 7 and 8 perform relative to the CRB?
- b. In line 184 of page 10, the authors state, "For moderate levels of wobbling, the CR lower bound for θ is essentially equal to that for θ_{3D} times $8\pi/3$." Based on Fig 4d, we can see $\theta_{3D} \approx 2$ for almost the entire range of θ . If both the statement and the figure are correct, then $\theta_{3D} \approx 5\pi$ everywhere. The cone angle θ must lie in the range $0 < \theta < 2\pi$. Therefore, having such a poor precision on measuring θ implies that CHIDO cannot resolve the difference between a fixed ($\theta = 0$) or freely rotating ($\theta = 2\pi$) emitter.
- i. Please verify the accuracy of the reported calculations.
 - ii. If accurate, the authors should comment on the poor performance for measuring wobble and how to improve it.
 - iii. Given these analyses, please also revise the following statements starting on line 212 appropriately: "These levels of precision are comparable or superior to those of other approaches..." In particular, "The determination of wobble is the most challenging, since (except for specific positions and orientations) significant accuracy might require the detection of tens of thousands of photons," is vague and presented without evidence.
 1. Ref. 4 measures wobble angle θ with 0.5-1 sr precision in solid angle using only ~ 400 photons. This techniques precision seems to be much better than that of CHIDO using far fewer photons.
 2. Ref. 12 measures the cone angle θ with 14° precision using 3000 photons.
 3. Ref. 17 measures cone angle θ with $\approx 9^\circ$ precision using ~ 1600 photons. Again, this seems to be much more efficient than the proposed method.
- c. The authors used fluorescent beads combined with polarizers to mimic in-plane and out-of-plane molecules in Figs. 5 and 6.
- i. Please quantify the accuracy of CHIDO for measuring wobble angle. That is, what is the theoretical expected wobble angle? What is the average estimated wobble angle from the experimental images? Please report these values in an SI Table.
 - ii. Please quantify the precision of CHIDO for measuring wobble angle. That is, compare the standard deviation of measured wobble angle to the CRB analysis.
 - iii. Compare the relative accuracy and precision of CHIDO. Is CHIDO's bias smaller than its precision for ~ 1000 detected photons? Or, is there a significant systematic bias present in the measurements?

- d. On line 332 of page 17, the authors reported the best expected precision for estimating wobbling angle. Please also report the mean or median precision of the measurements based on the experimental signal-to-background level.
3. More theoretical and experimental evidence is needed to show that the z position of fluorescent molecules can be measured accurately and precisely.
- a. The precisions for measuring the z position of nanobeads are reported to be ~25100 nm in Figs. 5,6 and Movies 2-4. However, “on the order of hundreds of thousands” (line 275) of photons were detected for each nanobead; therefore, the z precision should be ~1 nm (extrapolating from the ~10 nm precision for 1000 detected photons predicted by CRB in Fig. 4). Thus, CHIDO is performing >25x worse than expected for localization along z when measuring near-ideal fluorescent emitters.
Note that field-dependent aberrations, non-uniformity of the glass substrate, and variations in the size of nanobeads are systematic errors that affect each nanobead individually but should be constant for repeated measurements of the same bead. Therefore, these effects should principally degrade the accuracy, not precision, of the measurement.
Please perform a statistical analysis of the z accuracy and precision of CHIDO in this dataset, and report any discrepancies with respect to the CRB analysis.
- b. On line 322 of page 17, the authors attribute the large range of the estimated z positions of Alexa488-labeled F-actin (Fig. 8) to the non-planar deposition of filaments and tangling between filaments. However, the data in Fig. 8 indicate that molecules separated by ~100 nm laterally in x and y have z positions that differ by ~300 nm (e.g., green and red localizations next to each other, blue and red localizations near one another). The typical width of an actin filament is <10 nm.
- How can the z positions of labeled actin be so different over such a short distance, especially when they appear to be lying relatively flat on the coverslip?
 - These data suggest that the z measurements of these molecules are not accurate, not precise, or both. Please perform a statistical analysis of the z accuracy and precision of CHIDO in this dataset, and report any discrepancies with respect to the CRB analysis. I believe many more localizations need to be collected in order to obtain appropriate statistical power.
- c. These discrepancies in performance are a concern for users wanting to measure the 3D position and orientation of molecules: either the method is difficult to implement in a practical imaging system, or an analysis algorithm is difficult to write that achieves the theoretical performance depicted in Fig. 4, or both. The authors should quantitatively demonstrate that they are able to measure the z position of nanobeads within 2-5x of the CRB-expected precision, as is standard for comparable methods in the field.

4. Not much detail regarding the CR analysis is reported, and more information is needed in order to fully interpret the data.
 - a. Please give specific equations for exactly what is being calculated throughout the entire figure. The authors should state the number of background photons, measurement noise model (Poisson?), the emission wavelength, and numerical aperture of the imaging system used in the calculations.
 - b. Please compute the CR analysis for imaging conditions (i.e., signal and background) comparable to the experiments shown in Fig. 7 and 8.
 - c. Please clearly state how the inset of Fig. 4a was obtained.
 - i. What does “correlation in the parameters” mean? Does the inset directly depict the inverse of the Fisher information matrix?
 - ii. Are the correlations averaged over all x, y, z, β , considered in the paper? Or are they calculated for some specific values of these parameters? I believe they may vary dramatically for different positions and orientations.
 - iii. Why do the all diagonal terms equal to 1? These terms should be equal to CRB for $\beta = 0$, I believe.
 - d. I am confused by the x axis of Fig. 4b: shouldn't the z and precision degrade for small β (molecules oriented parallel to the optical axis), not large values?
 - e. The authors mention that they performed a robustness test with respect to image aberration by intentionally adding spherical aberration in their forward model (one wave, line 218 on page 12). Please quantitatively present the CR analysis in the SI for comparison.

Minor comment

There is no quantitative definition of “confidence” as used in the main paper, e.g., lines 247 and 263. I believe the authors are referring to eqn. (15) in SI.

1. Please explicitly define confidence in the main text, referring to the appropriate equation.
2. No specific examples are given to show how well this metric serves as a quantitative measure of goodness of fit. Please show some examples of molecules with “good” fits and “poor” fits from the data in Figs. 7 and 8.
3. Please give additional motivation for the merit/confidence functions given in SI eqns. (11) and (15). It seems to be quantifying mean-squared error, which is optimal for Gaussian-distributed noise. Is there a reason to choose this metric over a Poisson likelihood function?

Reviewer #3 (Remarks to the Author):

I have enjoyed reading the revised manuscript, which addressed all the points raised and in particular the experiment on densely labeled samples. I believe this is an excellent paper that will be valuable to many groups and strongly recommend its publication in Nature Communications.

Author's Response in Blue:

In this paper, Curcio, et al. propose a method to measure the 3D position, mean orientation, and wobble of fluorescent molecules. The authors used a stress-engineered optic (SEO) to modulate the electric field at the back focal plane of a microscope, followed by a Wollaston prism to separate right- and left-hand circularly polarized fluorescence. Termed Coordinate and Height super-resolution Imaging with Dithering and Orientation (CHIDO), the authors used Cramér-Rao bound calculations to predict the precisions of localizing single molecules in 3D, as well as measuring their orientation and wobble. They use polarization optics to modulate the fluorescence emission from beads in order to simulate dipole emitters and align/calibrate their imaging system. Finally, CHIDO is experimentally demonstrated by measuring the position and orientation of a collection of fluorophores attached to F-actin filaments.

The revised manuscript is significantly improved over the original version – the Cramér-Rao bound calculations show that the concept can theoretically achieve localization and orientation measurement precisions on par with existing techniques. However, my major concern lies with the experimental robustness of CHIDO. There is not enough evidence to conclude that the method can perform well, especially in terms of the accuracy and precision of measuring fluorophore z position and wobbling angle, in single-molecule super-resolution imaging. Measuring 3D position and wobble simultaneously are the principal innovations of the proposed approach, but they are not demonstrated convincingly. Detailed recommendations to address this concern and others are below.

Once more, we are very thankful to the reviewer for an extremely thorough reading of our manuscript and supplemental materials. The reviewer comments made us clarify further the message of our work, address accuracy issues in addition to precision. We also realized several errors that we accidentally introduced in the previous revision.

Major comments

1. Repeatedly throughout the manuscript, the authors write “accuracy” inappropriately when they should use the term “precision.” Examples include lines 206, 208, 210, 212, 332, 333, and 363. Measurement accuracy is the bias or error in the measurement when noise is absent or infinitesimal. Statistical precision is the variation or standard deviation of a set of repeated measurements taken under identical conditions. These two errors are separate

but are both important to quantify individually. Moreover, Cramér-Rao bound (CRB) analysis assumes that an unbiased (accurate) estimator is used, and thus predicts the best-possible variance (precision) of such an estimator. The CRB cannot be used to prove that a technique is accurate. Further, one must show that his/her estimator is unbiased to infer that any CRB analysis applies to it. Please revise the manuscript to distinguish between these concepts appropriately.

There are two parts to this comment that we will address separately.

First, the reviewer is absolutely right in that we were not careful with the terminology, and used repeatedly the term “accuracy” when we were really referring to “precision”. This error has been addressed, and we are thankful to the reviewer for pointing it out.

Second, the reviewer correctly indicates that CRB analysis assumes an unbiased estimator, a topic on which we elaborate later. This and other comments from the reviewer discussed in what follow made us realize that perhaps it was unclear what constitutes the proposed method itself (CHIDO), and what constitutes the estimation of the retrieved parameters. In our view, the central contribution reported in this manuscript is the new optical method for the encoding of height and directional information in the PSFs, and this is what we refer to as “CHIDO”. The CRB analysis provides meaningful estimations of the level of precision that this method can achieve. On the other hand, to treat the experimental data, we developed approximate techniques that were used for computational simplicity or to circumvent current experimental limitations, such as reference PSF models from nanobeads or the merit functions that are maximized. These estimators are not central to the method, and will be the focus of further work in the near future.

The issues with accuracy of the particular results presented depend more on limitations on the PSF models and the estimators used here, which certainly have room for improvement. Nevertheless, to address this important issue, we included Monte Carlo simulations that allow studying the accuracy and precision inherent to the method. These simulations show that the reached precision is to within a factor of 2 to 5 of that obtained in the CRB calculations, and that there is no significant bias on all estimated parameters (x, y, z, i_0, f_l), even in the presence of background.

2. More theoretical and experimental evidence is needed to show that molecular wobbling/dithering can be measured accurately and precisely.
 - a. The authors report the Cramér-Rao bound (CRB) $\alpha P3D$ of measuring a generalized 3D Stokes parameter (Fig. 4) instead of directly characterizing of the uncertainty of

measuring the cone solid angle θ_0 (reported in Fig. 8) or wobble angle S (reported in Fig. 7). This disagreement between theoretical predicted performance and actual experimental measurements makes it difficult to evaluate how well the technique performs in actual imaging experiments.

i. Please calculate the CRB for measuring θ_0 or S and report these values in Fig 4.

Please note that one of the main concerns by the reviewer regards precision in wobble estimation, which seemed overly problematic. This was due to a somewhat embarrassing error in the previous version: the code for generating the CR lower bound a_{p3D} used units of percent in order to make the curve fall in a scale more similar to that of the other parameters (in degrees or nanometers), but this was not reflected in the text, which was edited weeks later. This caused the estimation of the precision for wobble to be exaggerated by a factor of 100. An extra error, indicated by the reviewer, contributed another factor of 2. Therefore, after the revision, we believe it is clear that there is no fundamental problem for CHIDO's estimation of wobble. We thank the reviewer for pointing out these issues, that caused us to notice these important errors that made the method look significantly less precise than it can be.

We would like to clarify that P_{3D} is not the generalized 3D Stokes parameter, but a 3D measure of degree of polarization. In fact, in this revised version we show that this measure is equivalent to the rotational mobility parameter Y_{3D} discussed in Ref. 34 under the assumption of isotropic wobbling around a main direction (namely, when the two smallest eigenvalues of the 3×3 matrix are equal). This connection is now made in the manuscript, and a link is given to a manuscript just posted on arXiv that provides geometrical interpretations for P_{3D} , Y_{3D} , and other related measures. As discussed in the new version, given the linear dependence of the PSFs on P_{3D} , the use of this parameter is desirable from a theoretical point of view. For example, the manuscript now includes an approximate estimate of the CR lower bounds for CHIDO, and the estimate for a_{p3D} turns out to be approximately independent of all parameters (except for photon number and SBR). Further, the mathematical form of these estimates suggests a global measure of directional precision (including both angles and wobble) referred to as a_{Di} . We believe these new theoretical results add value to the work presented in the manuscript and highlight the desirable properties of CHIDO.

Despite the mathematical convenience of P_{3D} (or equivalently Y_{3D}), we followed the reviewer's advice and replaced, in Fig. 4, the plots for a_{p3D} with plots for a_n . We prefer not to use S because a_0 diverges (not only for CHIDO but for any method) in the limit of no wobble.

- ii. For the signal and background level of the experiments in Figs. 7 and 8, what is the CRB-predicted precision of estimating Ω or ?

As the reviewer correctly indicates, we had not discussed the effect of the background level. This is now incorporated in all CR calculations and we show that the calculated bounds are not majorly affected by background: they all scale approximately as the square root of $1+2/SBR$.

- iii. How well does the estimator used on the data in Figs. 7 and 8 perform relative to the CRB?

The single molecules (Fig. 7) and STORM-like (Fig. 8) data are retrieved under different estimation conditions. In the case of Fig. 7, we now provide standard deviations in Table 1 that are compared to the CR bounds. In the case of Fig. 8, which used a slightly different PSF shapes due to a slight increase in the value of the SEO parameter c , we noticed that the use of experimental PSFs obtained from beads was too limiting, since it was not representative of punctual objects. In this case, a purely theoretical PSF model was used, which introduces its own set of limitations. This model does not reflect small errors in the optical system such as residual aberrations or slight misalignments of the SEO with respect to the pupil, and we believe this discrepancy causes some problems in the accuracy. This can be appreciated in the PSFs of some of the molecules in the new Fig. 8, where due system imperfections the measured PSFs the “U” shapes might be more intense at certain parts than the theoretical ones (see for example G), possibly causing a bias in the estimation of z and ξ . As mentioned in the concluding remarks, future work will be dedicated to obtaining more reliable reference PSFs.

- b. In line 184 of page 10, the authors state, “For moderate levels of wobbling, the CR lower bound for Ω is essentially equal to that for $.7$ times $8/3$.” Based on Fig 4d, we can see $\sigma_{\Omega} \approx 2$ for almost the entire range of $.7$. If both statement and the figure are correct, then $3 \approx 5$ everywhere. The cone angle Ω must lie in the range $0 \leq \Omega \leq 2$. Therefore, having such a poor precision on measuring Ω implies that CHIDO cannot resolve the difference between a fixed ($\Omega = 0$) or freely rotating ($\Omega = 2$) emitter.
 - i. Please verify the accuracy of the reported calculations.
 - ii. If accurate, the authors should comment on the poor performance for measuring wobble and how to improve it.

We thank the reviewer for pointing out this error. For moderate levels of wobbling, the proportionality constant between the lower bounds for Ω and Ω_{3D} is $4\pi/3$ and not $8\pi/3$. However, the more serious error was the factor of a hundred mentioned earlier, due to a discrepancy between the code and the interpretation of the results. That is, our previous comments were off by a factor of 200! These two important errors, which strongly undersold CHIDO's ability to estimate wobble, have now been corrected.

- iii. Given these analyses, please also revise the following statements starting on line 212 appropriately: "These levels of precision are comparable or superior to those of other approaches..." In particular, "The determination of wobble is the most challenging, since (except for specific positions and orientations) significant accuracy might require the detection of tens of thousands of photons," is vague and presented without evidence.

Indeed, these comments have now been revised, in particular the second, which was overly pessimistic.

1. Ref. 4 measures wobble angle Ω with 0.5-1 sr precision in solid angle using only ~ 400 photons. This technique's precision seems to be much better than that of CHIDO using far fewer photons.

According to the revised CR estimates, for 400 photons (no background) CHIDO can achieve precisions between 0.3 sr (for a highly wobbling molecule) to 0.9 sr (for a nonwobbling molecule). Further, Ref 4 relies on specific situations that are different from those of CHIDO, based on a high index mismatch medium/surface and a limited distance to the surface, so it is difficult to make comparisons even if the levels of precision are similar.

2. Ref. 12 measures the cone angle S with 14° precision using 3000 photons.

It is difficult to make comparisons of precisions based on S given the highly nonlinear map between this parameter and Ω_{3D} . However, if we assume that, on average, this precision amounts to a precision in solid angle of about 0.5 sr, this is also comparable to the precision of CHIDO. We note that this approach does not provide an estimate of height, however.

3. Ref. 17 measures cone angle S with $\approx 9^\circ$ precision using ~ 1600 photons. Again, this seems to be much more efficient than the proposed method.

As with the previous method, it is difficult to make comparisons, since the angle used in this reference is not a cone angle because the information is projected onto 2D. Further, the method uses sequential illumination (and is therefore less appropriate for STORM imaging) and does not recover other parameters recovered by CHIDO.

- c. The authors used fluorescent beads combined with polarizers to mimic in-plane and out-of-plane molecules in Figs. 5 and 6.
 - i. Please quantify the accuracy of CHIDO for measuring wobble angle. That is, what is the theoretical expected wobble angle δ ? What is the average estimated wobble angle δ from the experimental images? Please report these values in an SI Table.

As commented at the beginning of this document, the accuracy of these measurement do not reflect fundamental issues with the accuracy of CHIDO (as shown from Monte Carlo simulations), but rather of the fitting model used here. For the nanobeads, we used a PSF model based on a quadratic fit in z , whose motivation was mainly that of computational simplification when using experimental PSFs. To stress this fact, we moved the discussion on the polynomial model away from the main description of the method and into the section where the bead measurements are presented. The goal of the bead measurements was essentially to show that the method can estimate height and orientations. The measured PSFs include no wobble. A more informative indicator of the ability of CHIDO to retrieve wobble information is given by the results of the Monte Carlo simulations included in the new version of the Supplementary Materials, where the precision in all parameters (including wobble) is on the order of 2 to 5 times the CR bounds.

- ii. Please quantify the precision of CHIDO for measuring wobble angle. That is, compare the standard deviation of measured wobble angle δ to the CRB analysis.

For the same reason, because wobble is not naturally included in the sub-basis set of PSFs used in each of these measurements, we do not believe that such estimates would be useful, and we think that the Monte Carlo estimates are more informative.

- iii. Compare the relative accuracy and precision of CHIDO. Is CHIDO's bias smaller than its precision for ~ 1000 detected photons? Or, is there a significant systematic bias present in the measurements?

This point is now addressed in the Monte Carlo simulations added to the new version of the Supplementary Materials, where we found no inherent bias within the precision level.

- d. On line 332 of page 17, the authors reported the best expected precision for estimating wobbling angle. Please also report the mean or median precision of the measurements based on the experimental signal-to-background level.

This statement was based on the erroneous interpretation of the results, which underestimated the precision for estimating wobble by a factor of $\sqrt{200}$. The text has been changed and a representative range is now given (not only the best expected precision).

3. More theoretical and experimental evidence is needed to show that the z position of fluorescent molecules can be measured accurately and precisely.
 - a. The precisions for measuring the z position of nanobeads are reported to be $\sqrt{25-100}$ nm in Figs. 5,6 and Movies 2-4. However, “on the order of hundreds of thousands” (line 275) of photons were detected for each nanobead; therefore, the z precision should be $\sqrt{1}$ nm (extrapolating from the $\sqrt{10}$ nm precision for 1000 detected photons predicted by CRB in Fig. 4). Thus, CHIDO is performing $>25\times$ worse than expected for localization along z when measuring near-ideal fluorescent emitters. Note that field-dependent aberrations, non-uniformity of the glass substrate, and variations in the size of nanobeads are systematic errors that affect each nanobead individually but should be constant for repeated measurements of the same bead. Therefore, these effects should principally degrade the accuracy, not precision, of the measurement. Please perform a statistical analysis of the z accuracy and precision of CHIDO in this dataset, and report any discrepancies with respect to the CRB analysis.

We made significant changes to this part of the manuscript in response to the reviewer comments. As the reviewer mentions, glass-substrate variations would cause errors in the height estimates that are systematic. On the other hand, we believe that the effect of field-dependent aberrations would make the reference PSFs vary with xy position, and there is no reason to expect that the effect of these variations on height estimation would be just a shift. Regarding bead size variations, their effect is not only to lift the center of the bead, but as is now explained in more detail in the manuscript, the beads are sufficiently large as to cause appreciable blurring in the PSFs and hence removing some of their features. Different levels of blurring will result from different sizes, and the effect on height estimation will probably not be just a simple systematic offset. Nevertheless, we believe that the suggestion by the reviewer of looking at z increments for each bead to remove systematic error is good, so we selected the group of beads for which the measured PSFs provided good fits over all heights, and estimated the standard deviations of these height increments. This procedure did result in a reduction of the spread of about 10-20%.

As just mentioned, the spatial extension of the beads causes a blurring of fine details of the PSFs and this blurring affects also the CR bounds. Therefore, in order to make the comparison of experimental results with theory, we calculated the CR bounds in the estimation of height not from the theoretical model used for the generation of Fig. 4 (which is more representative of single molecules) but from the model constructed from the beads themselves. This blurring effect was found to account for a factor of a bit less than 2. In addition, as the reviewer pointed out, we had not included the effect of background, and the SBR for the bead measurements is of the order 1/2 to 5, since there was significant variation of photon number, appreciable from the figures and the movies. Once these effects are taken into account, the CR lower bound obtained from the bead-based PSF model for SBR = 3 and 50000 photons (given that some PSFs have significantly less photons than others), the discrepancy between the measured standard deviations and the CR lower bound is only of a factor of 4 to 5, both when the s-wave plate and the linear polarizer were used. This remaining discrepancy is reasonable, and it is at least partly due, as is now mentioned in the manuscript, to imperfections in the model and to non-systematic contributions from aberrations and bead size variations.

- b. On line 322 of page 17, the authors attribute the large range of the estimated z positions of Alexa488-labeled F-actin (Fig. 8) to the non-planar deposition of filaments and tangling between filaments. However, the data in Fig. 8 indicate that molecules separated by ~100 nm laterally in x and y have z positions that differ by ~300 nm (e.g., green and red localizations next to each other, blue and red localizations near one another). The typical width of an actin filament is <10 nm.
 - i. How can the z positions of labeled actin be so different over such a short distance, especially when they appear to be lying relatively flat on the coverslip?

Prompted by this comment by the reviewer, we analyzed these results more carefully. First, an unfortunate error in the STORM analysis program estimated height in units of wavelengths but failed to account for the refractive index of the medium in which the filaments are embedded. (Please note that the effects of this refractive index on the transmission coefficients between media had been taken into account.) Therefore, the distances were exaggerated by a factor equal to this refractive index. This is now corrected. Second, we now concentrate the analysis on data that exceeds a given threshold of confidence level (namely, the normalized correlation between the measured PSFs and the model) and found that this level is not necessarily related to the number of photons. Finally, we note that the use of a theoretical PSFs model instead of an experimental one leads to (probably systematic) errors. As mentioned earlier and clarified in the manuscript, the construction of an experimentally-based PSFs basis is not obvious and requires further work. The use of

purely theoretical PSFs in this context was motivated by the inadequacy of the nanobeads to construct the PSF basis, not only because their size causes blurring, but also due to the incompleteness of the resulting basis. The theoretical PSF basis fails, however, in accounting for small residual aberrations and other setup imperfections, which affects the accuracy of the results. By considering only those fluorophores for which the correlation between theory and measurement is more than 0.35 (this level of confidence being now encoded in the size of the cones), the strong jumps in height between neighboring fluorophores largely disappear, at the cost of course of seeing less fluorophores along the filaments. The new version of Fig. 8 now accounts for this confidence level filter. Note that we only retained one of the three zoomed sections of the previous versions, because it is the one where more molecules were detected with high confidence and because it shows two nearby molecules attached to different filaments. We also considered that it was more interesting to show in the rest of the figure all the measured PSFs for this region, as well as their theoretical matches.

While more measurements would be desirable, these are difficult to perform in the current situation, so we addressed the reviewer comments by analyzing further the data we had obtained. We do believe, however, that the results presented in the manuscript, supplemented with Monte Carlo simulations in conditions similar to the experiments, in combination with the Cram r Rao theoretical calculations, give sufficient evidence of the potential of CHIDO in STORM conditions. As is clarified in the manuscript, the next step is to develop methods for obtaining more reliable PSF models, probably through a mix of theoretical and experimental techniques, which will improve the accuracy of the method.

- c. These discrepancies in performance are a concern for users wanting to measure the 3D position and orientation of molecules: either the method is difficult to implement in a practical imaging system, or an analysis algorithm is difficult to write that achieves the theoretical performance depicted in Fig. 4, or both. The authors should quantitatively demonstrate that they are able to measure the z position of nanobeads within 2-5x of the CRB-expected precision, as is standard for comparable methods in the field.

With the revisions to the calculations and the analysis of the data, we show that experimental data approach theoretical expectations, with measurement precision within a factor of 5 from the CR bounds calculated from the appropriate model. The nanobead demonstration gives in our opinion ample evidence of the capacity of the method to be implemented and to provide results with high precision and accuracy. Concerning single molecule measurements, a more reliable PSF basis is indeed required, and this will be the focus of our research in the near future. Let us stress that the main point of the current work

is to demonstrate the principle of CHIDO, which is in itself a breakthrough concept in the context of 3D position/orientation super resolution imaging, and to provide sufficient elements to show that it works, which we believe we do. As is the case with other experimental techniques, further refinements will follow.

4. Not much detail regarding the CR analysis is reported, and more information is needed in order to fully interpret the data.

- a. Please give specific equations for exactly what is being calculated throughout the entire figure. The authors should state the number of background photons, measurement noise model (Poisson?), the emission wavelength, and numerical aperture of the imaging system used in the calculations.

This information has now been included in the Supplementary Materials, including the derivation of simple estimates that are given in the main manuscript. Plots were added that reflect the number of background photons (assuming Poisson noise) that is typical of the single molecule measurements presented. The emission wavelength (520 nm) and numerical aperture (1.45 oil immersion) are described in the document.

- b. Please compute the CR analysis for imaging conditions (i.e., signal and background) comparable to the experiments shown in Fig. 7 and 8.

This has been done. CR result with SBR comparable to that of the bead measurements are now shown within the right column of Fig. 4. While the plots in Fig. 4 are given for 10 000 photons, numbers concerning single molecules are given in the single molecules sections.

- c. Please clearly state how the inset of Fig. 4a was obtained.
 - i. What does “correlation in the parameters” mean? Does the inset directly depict the inverse of the Fisher information matrix?

The definition of a correlation implies normalization by the square root of the product of the corresponding diagonal values. This is now explained in the Supplementary Materials. Please note, however, that we removed this inset, and instead include figures that are averages and standard deviations over all parameters in the supplementary materials, for different values of SBR.

- ii. Are the correlations averaged over all x , y , z , δ , considered in the paper? Or are they calculated for some specific values of these parameters? I believe they may

vary dramatically for different positions and orientations.

The new correlations are averaged over all these parameters, as is now described in the Supplementary Materials. Further, we now show both the average and standard deviations of these correlations, for two values of the SBR.

iii. Why do the all diagonal terms equal to 1? These terms should be equal to CRB for $\delta = 0$, I believe.

As is now explained in the Supplementary Materials, these correlations are normalized, so the diagonals are unity (full correlation).

d. I am confused by the x axis of Fig. 4b: shouldn't the z and precision degrade for small δ (molecules oriented parallel to the optical axis), not large values?

The reviewer is absolutely right. There was an error in the labeling of this axis. In any case, a new version of the figure was generated.

e. The authors mention that they performed a robustness test with respect to image aberration by intentionally adding spherical aberration in their forward model (one wave, line 218 on page 12). Please quantitatively present the CR analysis in the SI for comparison.

The plots are now shown in the Supplementary Materials.

Minor comments

There is no quantitative definition of "confidence" as used in the main paper, e.g., lines 247 and 263. I believe the authors are referring to eqn. (15) in SI.

1. Please explicitly define confidence in the main text, referring to the appropriate equation. Confidence is the normalized correlation of the measured PSF and the model PSF for the retrieved parameters. This is now explained in the manuscript.
2. No specific examples are given to show how well this metric serves as a quantitative measure of goodness of fit. Please show some examples of molecules with "good" fits and "poor" fits from the data in Figs. 7 and 8.

The new version of Fig. 8 shows the measured PSFs together with the theoretical fits, and specifies the numerical value of the correlation.

3. Please give additional motivation for the merit/confidence functions given in SI eqns. (11) and (15). It seems to be quantifying mean-squared error, which is optimal for Gaussian-distributed noise. Is there a reason to choose this metric over a Poisson likelihood function?
We agree that likelihood is more suitable given Poisson noise. However, the decomposition into a basis is closer to a mean-squared error. As shown by the Monte Carlo simulations, the results are acceptable. However, in future work we will indeed most likely use maximum likelihood estimation. Let us once more stress that the chosen merit function to maximize is not an integral part of the proposed technique, but one that was chosen for convenience.

Reviewer #2 (Remarks to the Author):

“Birefringent Fourier filtering for single molecule Coordinate and Height super-resolution Imaging with Dithering and Orientation (CHIDO)”

In this paper, Curcio, et al. propose a method to measure the 3D position, mean orientation, and wobble of fluorescent molecules. The authors used a stress-engineered optic (SEO) to modulate the electric field at the back focal plane of a microscope, followed by a Wollaston prism to separate right- and left-hand circularly polarized fluorescence. Termed Coordinate and Height super-resolution Imaging with Dithering and Orientation (CHIDO), the resulting PSF largely orthogonalizes molecular 3D position, 3D orientation, and wobble such that they are distinguishable on an emCCD camera. Cramér-Rao bound calculations show that the CHIDO PSFs are reasonably robust to background photons for estimating 3D position, 3D orientation, and molecular wobble. The authors use polarization optics to modulate the fluorescence emission from beads to simulate dipole emitters and align/calibrate their imaging system. Finally, CHIDO is experimentally demonstrated by measuring the position and orientation of a collection of fluorophores attached to F-actin filaments.

This reviewer finds this revised manuscript to be much improved over previous versions. In particular, I appreciate the authors' care in providing more details surrounding the method and its characterization, advantages and weaknesses of the method, and a thorough response to issues identified during the previous review. I believe the manuscript will be suitable for publication in *Nature Communications* after a few minor comments, detailed below, are addressed.

1. No details about the fabrication of the stressed-engineered optic (SEO) at the heart of the CHIDO method are provided in this manuscript, aside from some references to previous literature. Please give a basic description in the Methods section of how to fabricate or obtain the SEO.
2. I suggest using script fonts or different notation to describe left hand circular (LHC) and right hand circular (RHC) polarized light at the detector (line 106). Currently, it is very easy to misread “ $p = l$ ” (letter L) as “ $p = 1$ ” (numeral 1), e.g., in the caption of Fig. 2.
3. Please state the equivalent background photon flux, in photons per area, for $SBR = 1/3$ in the caption of Fig. 4.
Signal to background ratio (SBR) is defined on line 174 as the ratio of the peak point spread function (PSF) intensity to the background photon intensity. However, since an integrated signal of “10000 photons” is given in this figure caption, it is difficult to calculate or interpret “ $SBR = 1/3$ ” in the context of a typical single-molecule experiment.

Author's response in **Blue**:

In this paper, Curcio, et al. propose a method to measure the 3D position, mean orientation, and wobble of fluorescent molecules. The authors used a stress-engineered optic (SEO) to modulate the electric field at the back focal plane of a microscope, followed by a Wollaston prism to separate right- and left-hand circularly polarized fluorescence. Termed Coordinate and Height super-resolution Imaging with Dithering and Orientation (CHIDO), the resulting PSF largely orthogonalizes molecular 3D position, 3D orientation, and wobble such that they are distinguishable on an emCCD camera. Cramér-Rao bound calculations show that the CHIDO PSFs are reasonably robust to background photons for estimating 3D position, 3D orientation, and molecular wobble. The authors use polarization optics to modulate the fluorescence emission from beads to simulate dipole emitters and align/calibrate their imaging system. Finally, CHIDO is experimentally demonstrated by measuring the position and orientation of a collection of fluorophores attached to F-actin filaments.

This reviewer finds this revised manuscript to be much improved over previous versions. In particular, I appreciate the authors' care in providing more details surrounding the method and its characterization, advantages and weaknesses of the method, and a thorough response to issues identified during the previous review. I believe the manuscript will be suitable for publication in Nature Communications after a few minor comments, detailed below, are addressed.

We are delighted to hear that the reviewer found that the revisions improved significantly the quality of the manuscript. We totally agree, and we thank the reviewer for all the useful feedback. As described in what follows, we have now implemented the last three suggestions by the reviewer.

1. No details about the fabrication of the stressed-engineered optic (SEO) at the heart of the CHIDO method are provided in this manuscript, aside from some references to previous literature. Please give a basic description in the Methods section of how to fabricate or obtain the SEO.

A brief description of the fabrication process for the SEO is now provided in the Methods section.

2. I suggest using script fonts or different notation to describe left hand circular (LHC) and right hand circular (RHC) polarized light at the detector (line 106). Currently, it is very easy to misread "p = l" (letter L) as "p = 1" (numeral 1), e.g., in the caption of Fig. 2.

We are very happy that the reviewer caught this possible source of confusion. We now use script, capital letters to label the values of p indicating left- and right-circular polarization.

3. Please state the equivalent background photon flux, in photons per area, for $SBR = 1/3$ in the caption of Fig. 4. Signal to background ratio (SBR) is defined on line 174 as the ratio of the peak point spread function (PSF) intensity to the background photon intensity. However, since an integrated signal of "10000 photons" is given in this figure caption, it is difficult to calculate or interpret " $SBR = 1/3$ " in the context of a typical single-molecule experiment.

This is also a good suggestion that we now have incorporated into the caption of Fig.

0. The corresponding quantity is about 250 photons per pixel.